# ABE-ultramax for high-efficiency biallelic adenine base editing in zebrafish

Wei Qin [1,6], Fang Liang [2,6], Sheng-Jia Lin [1], Cassidy Petree[1], Kevin Huang [1], Yu Zhang[1], Lin Li[3,4], Pratishtha Varshney[1], Philippe Mourrain[5], Yanmei Liu [3,4] ✉ & Gaurav K. Varshney [1] ✉

Advancements in CRISPR technology, particularly the development of base editors, revolutionize genetic variant research. When combined with model organisms like zebrafish, base editors significantly accelerate and refine in vivo analysis of genetic variations. However, base editors are restricted by proto-spacer adjacent motif (PAM) sequences and specific editing windows, hindering their applicability to a broad spectrum of genetic variants. Additionally, base editors can introduce unintended mutations and often exhibit reduced efficiency in living organisms compared to cultured cell lines. Here, we engineer a suite of adenine base editors (ABEs) called ABE-Ultramax (Umax), demonstrating high editing efficiency and low rates of insertions and deletions (indels) in zebrafish. The ABE-Umax suite of editors includes ABEs with shifted, narrowed, or broadened editing windows, reduced bystander mutation frequency, and highly flexible PAM sequence requirements. These advancements have the potential to address previous challenges in disease modeling and advance gene therapy applications.

Around 90% of all known human disease-causing genetic variants can be attributed to single-nucleotide variants (SNVs)[1]. However, nearly half of all SNVs are still classified as variants of unknown significance (VUS)[2]. Functional studies in vertebrate model organisms are essential to determine the biological impact of these variants and their potential to cause disease[3]. A key enabling technology in this area is CRISPR genome editing[4], but the precise installation of specific single-nucleotide changes at desired positions in model organism genomes remains a challenge. Base editors (BEs) enable the introduction of point mutations at targeted genomic sites with higher efficiency and precision than traditional cleavage-based genome editing methods[5]. Cytosine base editors (CBEs) consist of a Cas9 nickase (nCas9) fused with a cytosine deaminase enzyme, enabling specific C•G to T•A conversions and often paired with a uracil glycosylase inhibitor (UGI)[5]. In

contrast, adenine base editors (ABEs) fuse nCas9 to an adenine dea-minase (TadA or eTadA) to accomplish A•T to G•C alterations[6].

In CRISPR-based editors including ABEs, Cas9 complexed with a single guide RNA (sgRNA) forms an R-loop structure at the target DNA site. In ABEs specifically, adenine deaminase fused with Cas9 nickase converts the exposed adenine to inosine. During DNA replication, inosine is misread as guanosine, resulting in a permanent A-to-G base change. Additionally, Cas9 nickase (nCas9) introduces a nick in the complementary strand, prompting the DNA repair machinery to pre-ferentially use the edited strand as a template, further boosting editing efficiency[6]. ABE8e, the most widely used ABE variant due to its effi-ciency and compatibility with a wide range of model systems[7–10], typically exhibits an editing window spanning positions 4–8 relative to the protospacer adjacent motif (PAM) sequence on the target DNA

[1]Genes & Human Disease Research Program, Oklahoma Medical Research Foundation, Oklahoma City, OK, USA. [2]Institute of Modern Aquaculture Science and Engineering, School of Life Sciences, South China Normal University, Guangzhou, Guangdong 510631, China. [3]Key Laboratory of Brain, Cognition and Education Sciences, Ministry of Education, South China Normal University, 510631 Guangzhou, China. [4]Institute for Brain Research and Rehabilitation, and Guangdong Key Laboratory of Mental Health and Cognitive Science, South China Normal University, 510631 Guangzhou, China. [5]Department of Psychiatry and Behavioral Sciences, Stanford University, Stanford, CA, USA. [6]These authors contributed equally: Wei Qin, Fang Liang. ✉e-mail: yanmeiliu@m.scnu.edu.cn; gaurav-varshney@omrf.org

strand. However, its targeting efficiency remains low at some specific sites, and its constrained editing window and preference for NGG PAM severely limit the sequence space that can be targeted by existing base editors. Recent studies have shown improved editing efficiencies in human cells using variants of the TadA deaminase domain (e.g., TadA-KR and TadA-9e). However, their in vivo efficiencies remain limited or have not been tested[11,12]. Additionally, ABE tools with activity in shifted editing windows have been developed[13,14]. Furthermore, researchers have engineered Cas9 variants with superior efficiency, fidelity, and diverse PAM preferences[15–18]. Notably, the SpRY variant recognizes a broader "NNN" PAM and demonstrates efficient, near PAM-independent genome editing in zebrafish[19].

Approximately 47% of disease-causing SNVs involve the substitution of G(C) to A(T)[1], underscoring the huge potential of ABEs, as they could be used to correct nearly half of all pathogenic SNVs. However, current ABE platforms are limited by low editing efficiencies in vivo, restricted targeting potential due to protospacer adjacent motif (PAM) requirements and size of the editing window, and undesirable formation of insertions and deletions (indels) at the target site[7,20,21]. Next-generation ABEs have been optimized for high activity in human cell lines through directed evolution or protein engineering[11,12,22,23], but still suffer from reduced in vivo activity, targeting limitations, and bystander mutations. Therefore, there remains a significant unmet need for ABEs optimized for high in vivo editing efficiency with low rates of indels and bystander mutations, expanded PAM recognition for higher versatility, and superior editing precision.

Here, we report the development and characterization of ABE-Ultramax, an ABE platform with high efficiency and specificity for biallelic in vivo adenine base editing. We develop ABE-Ultramax variants with shifted, narrowed, or widened editing windows, relaxed PAM preference, and germline-specific targeting ability and demonstrate their ability to establish causality for coding and non-coding human pathogenic SNVs in the zebrafish model system.

## Results

### Screening for an ABE effector for efficient in vivo editing

Many TadA variants have been developed by various research groups utilizing diverse strategies to introduce mutations in TadA enzyme[11,12,22,23]. We reasoned that combining individual TadA and Cas9 variants may synergistically improve ABE efficiency and specificity. To test this hypothesis, we developed ten TadA variants using different permutations of previously validated mutations (Supplementary Fig. 1). As a model for evaluating the in vivo transition activity of these ABE variants, we used albino/nacre *mitfa*[W2/W2] zebrafish, which harbor a C-to-T point mutation at position 417 of the *mitfa* gene resulting in impaired melanin production and in turn in a white fish body phenotype. A•T to G•C conversion at this site (targeting the antisense adenine at position 417) restores melanin production, resulting in a visual black skin pigmentation recovery phenotype (Fig. 1a). Since the target adenine was not within the targeting window for any protospacer with the "NGG" PAM, we fused our TadA variants with the PAM-less Cas9 variant known as SpRY Cas9[18]. We injected gRNAs targeting adenine 417 and mRNAs encoding one of our ten ABE variants into homozygous *mitfa*[W2/W2] embryos and classified them into four bins based on their rate of pigmentation recovery: Low rescue means embryos have 1–10 skin melanin-expressing cells/melanocytes; medium rescue means 10–40 melanocytes and high rescue means over 40 melanocytes, similar to wild-type (WT) embryos (Fig. 1b). We observed a range of activities among the ten TadA variants, with the double mutant V82S/Q154R, hereafter referred to ABE-Ultramax-SpRY (or ABE-Umax-SpRY for short) performing the best out of the ten variants in terms of their ability to rescue pigmentation phenotype. Editing with ABE-Umax-SpRY achieved high rescue in 26% of embryos, medium rescue in 30%, and low rescue in 38% of embryos (Fig. 1c). High-throughput

sequencing verified that T to C correction was responsible for the rescued phenotype (Supplementary Fig. 2).

To quantify the impact of the TadA variant on base editor performance, we compared the targeting efficiencies between ABE-Umax-SpRY and ABE8e-SpRY at nine reported loci[19]. To streamline and expedite the evaluation of base conversion efficiency, we utilized the established EditR software for quantitative analysis of Sanger sequencing data. This approach is described in previous publications[19,24,25]. Our results demonstrated that ABE-Umax-SpRY can target all sites effectively, exhibiting editing efficiencies up to 86% (Fig. 1d), representing a threefold improvement in the mean editing efficiency over the current state-of-the-art ABE8e-SpRY variant in vivo (Supplementary Fig. 3). Interestingly, ABE-Umax-SpRY can target sites such as *rpl9-NTC* that could not be previously edited by ABE8e-SpRY[19], indicating the critical combination of a high-efficiency editor with a broadened PAM preference for editing difficult to edit sites in zebrafish.

Previous reports have uncovered strong locus preference among existing ABEs; for example, the ABE7.10 variant exhibited less than 25% editing efficiency at only 5 out of 28 tested loci, despite the presence of favorable PAM sequences and adenine positioning in the editing window[20]. To assess whether this issue persists with our developed variant, we characterized the editing profile of ABE-Umax-SpRY across 16 endogenous zebrafish target sites spanning 11 genes. We selected target sites with diverse PAM sequences (NRN and NYN) in previously tested[19] and other genes not previously tested. Notably, we detected base editing at all 16 sites, with 13 sites demonstrating editing efficiency above 51.33% (the maximum efficiency within the editing window) (Fig. 1e), demonstrating that our variant minimizes the locus-dependent issues that have generally hampered previous base editors. Upon analyzing all editing sites with NNN PAM in this study, we determined that the editing window of this tool ranges from positions 3 to 9, with heightened activity notably observed at positions 4 to 8 on the protospacer (Fig. 1f).

The SpRY Cas9 variant is known to exhibit weak recognition towards the canonical "NGG" PAM, and while its relaxed PAM preference can enable targeting of additional loci, it also can produce potential off-target editing due to lower specificity[18]. To mitigate these potential issues for sites that can be targeted with an "NGG" PAM, we replaced the Cas9 moiety in ABE-Umax-SpRY with the canonical SpCas9, creating ABE-Ultramax (ABE-Umax) and compared its editing efficiency at eight reported genomic sites harboring an "NGG" PAM sequence[20,26]. ABE-Umax indeed outperformed ABE-Umax-SpRY at loci with "NGG" PAM with a 2.09-fold improvement in mean editing efficiency (Fig. 2a, b). Furthermore, ABE-Umax exhibited robust biallelic editing performance across the same loci (Fig. 2a) achieving over 90% editing efficiency in some cases, with a 1.42-fold increase in mean transition editing efficiency in comparison to ABE8e (Fig. 2b), which is currently the most efficient ABE available.

To ensure our optimized tool can edit a diverse set of loci implicated in human disease, we further characterized the editing profile of ABE-Umax across an additional 28 endogenous zebrafish target sites with NGG PAM spanning 17 genes of interest. All tested sites exhibited high editing efficiency (mean efficiency 57.06%, up to almost 100% at five sites) within the main editing window (positions 4–8 on the protospacer) with the expected A-to-G transition as the predominant product (Fig. 2b). Sanger sequencing did not reveal indels or unwanted editing around the locus at the above target sites. However, Sanger sequencing has limited ability to resolve unwanted changes with efficiencies below 5%, so we used next-generation sequencing (Miseq) to deeply profile three of these sites as a more sensitive measure of indels and bystander editing. The results of the Miseq sequencing analysis were consistent with the editing efficiencies initially determined by Sanger sequencing (Supplementary Fig. 4). Meanwhile, ABE-Umax exhibited high product purity, with only A-to-G transitions being

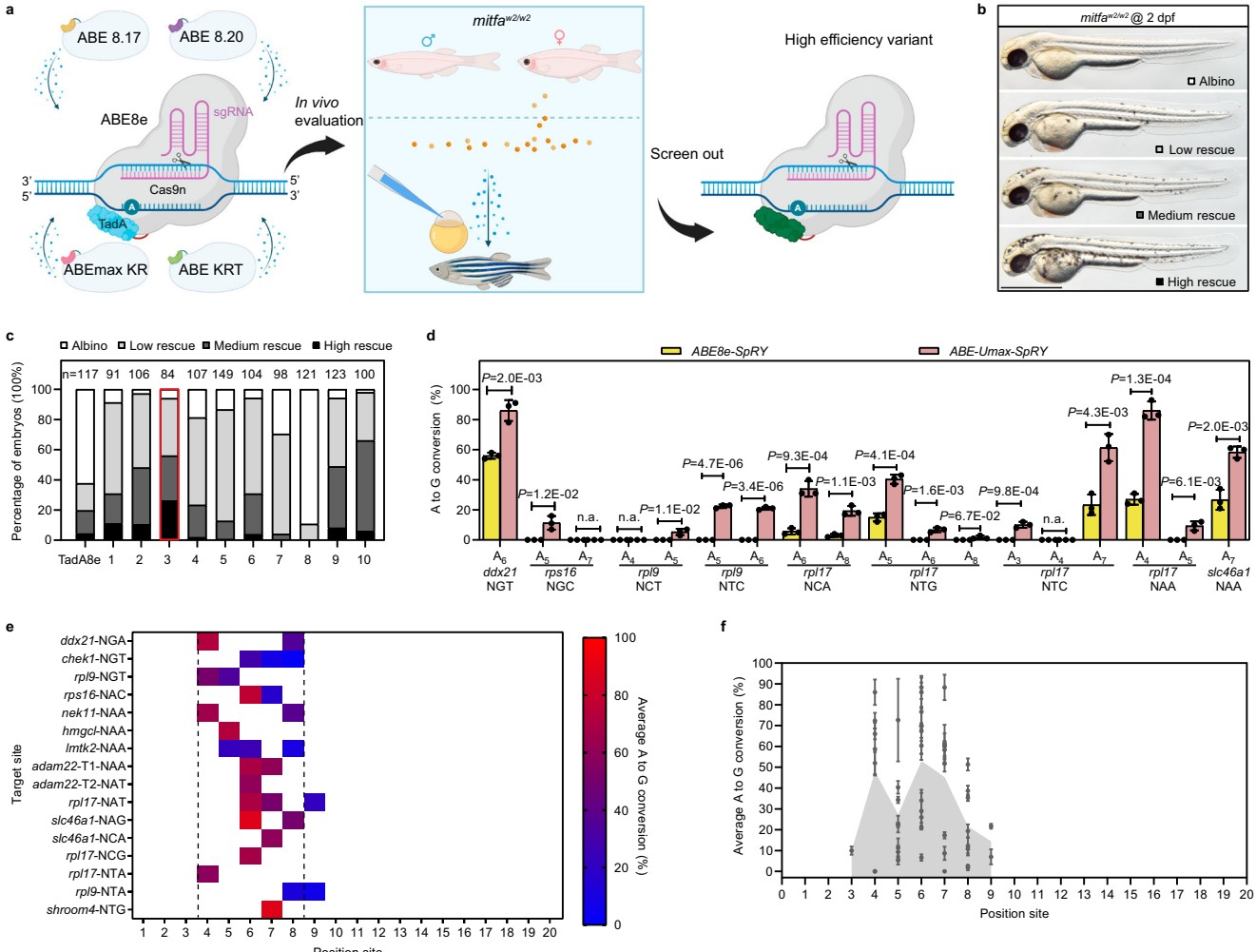

**Fig. 1 | Efficient adenine base editing mediated by ABE-Umax-SpRY. a** Schematic diagram of the screening process for efficient adenine base editing tools. Cas9n Cas nickase, TadA adenine deaminase, PAM protospacer adjacent motif, sgRNA single guide RNA. Different colors (blue, yellow, purple, pink, and green) represent various TadA variants used in the deaminases. Created with BioRender.com released under a Creative Commons Attribution-NonCommercial-NoDerivs 4.0 International license. **b** Lateral views of embryos at 2 days post-fertilization (dpf) are displayed. The degree of mosaic pigmentation compared to the mutant is categorized. Scale bar: 1 mm. **c** A comparison of the efficiency of various adenine base editing tools in correcting point mutations in the *mitfa^{W2/W2}* mutant line. The stacked columns display the percentages of albino (white), low rescue (light gray), medium rescue (dark gray), and high rescue (black) embryos. The count of embryos is displayed above each column. The most efficient tool, ABE-Umax-SpRY, is highlighted in a red box. **d** Editing efficiency comparison between ABE8e-SpRY and ABE-Umax-SpRY using nine gRNAs targeting NNN PAMs. The edited base position within each gRNA is indicated by numbers. Data were presented as mean ± standard deviation (SD) calculated from three biological replicates. Two-tailed paired *t*-test were performed (with *P* values marked). **e** The heatmap illustrates the mean A-to-G editing efficiency of ABE-Umax-SpRY across 16 target sites at various positions within the protospacer. Editing efficiency (A-to-G conversion) is depicted using a color gradient, ranging from red (100%) to blue (0%). A-to-G conversion frequencies for each adenine nucleotide within the 20-bp protospacer were quantified using EditR. The heatmap displays conversion frequencies exceeding 0.20, averaged across three independent experiments. The black dotted line delineates ABE-Umax-SpRY's preferred editing window, encompassing positions 4 through 8. **f** Assessment of the efficiency and targeting window of ABE-Umax-SpRY. Each data point represents the average editing activity ± standard deviation (SD) at a particular site based on the data in Fig. 1d, e. The targeting window of ABE-Umax-SpRY is colored in gray from position 3 to 9, counting from the 5′ terminal to the 3′ terminal of the targeting site. All source data in this figure are provided as a source data file.

observed at the three sites. The indel frequency was also very low, ranging from 0.50 to 4.73% across various sites, compared to the previously reported 7.14 to 22.20% indels range for ABE7.10 in zebrafish[20].

The above analyses demonstrated that we could achieve high levels of somatic editing in injected embryos (F0), which is useful for rapid phenotypic screening. However, stable genetic knockout lines that can produce edited progeny (F1 and beyond) are typically needed for mechanistic and functional studies of disease mutations. Here, we investigated zebrafish germline targeting efficiencies of ABE-Umax for a total of eight targets (from 28 sites tested), showing that all target sites exhibited more than 50% editing efficiency, with seven of eight targets achieving more than 65% germline editing efficiency, including two targets having an 80%, and one site with striking 100% editing efficiency (Supplementary Data 1).

Biallelic mutations in many genes, such as those related to development, can lead to early lethality in zebrafish, making it challenging to generate stable genetic lines for phenotypic analysis and/or to cross different mutations into the same progeny[27]. To overcome this hurdle, we wanted to develop a variant that would only generate mutations in the germline, thereby maintaining the viability of the F0 animals when targeting essential genes. In zebrafish, the 3′ UTR of *nanos1* controls its exclusive germline expression pattern[28], and previous studies have shown that this 3′ UTR can direct germline-specific

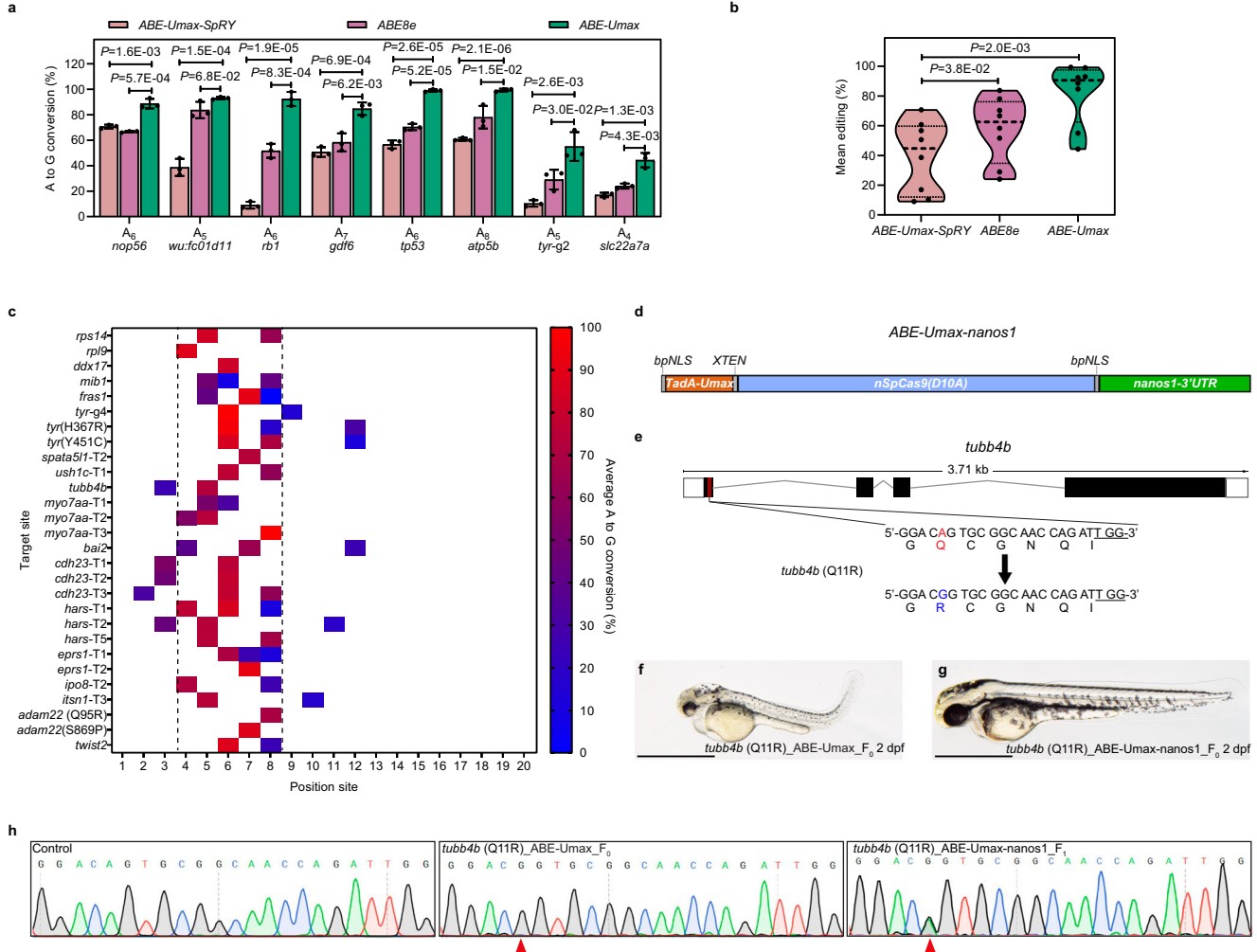

**Fig. 2 | Efficient adenine base editing mediated by ABE-Umax. a** The editing efficiency comparison among ABE-Umax-SpRY, ABE8e-SpRY, and ABE-Umax targeting eight different loci with NGG PAM. The editing base position within the gRNA is indicated by numbers. Values are presented as mean value ± standard deviation (SD), *n* = 3 biological replicates. Two-tailed paired *t*-test were performed (with *P* values marked). **b** Assessment of the mean editing efficiency of ABE-Umax-SpRY, ABE-Umax, and ABE8e Umax using the plot based on the data in Fig. 2a. Mean editing activity at each site is represented by individual data points, with the central dotted line indicating the overall mean. Two-tailed paired *t*-test were performed (with *P* values marked at the top of the violin plot.). **c** Heatmap depicting the average A-to-G editing efficiency of ABE-Umax across 28 target sites. Editing efficiency is represented by a color gradient, with red indicating 100% efficiency and blue indicating 0% efficiency. The black dotted line highlights the preferred editable range of ABE-Umax, between positions 4 and 8. **d** The mRNA construct of ABE-Umax-nanos1 for germline-specific adenine base editing. **e** Schematic diagram of the *tubb4b* locus. The targeted sequence is shown with the PAM underlined. The targeted nucleotide and amino acid are highlighted in red, while the expected changes in nucleotide and amino acid are highlighted in blue. **f, g** Phenotype of F0 embryos injected with *tubb4b* gRNA and ABE-Umax mRNA (**f**) or ABE-Umax-nanos1 mRNA (**g**). Lateral views of 2 dpf embryos are shown. Scale bar: 1 mm. **h** Sanger sequencing chromatograms confirm the A to G editing events in F0 injected embryos and F1. The desired mutation is indicated by a red arrowhead. All source data in this figure are provided as a source data file.

expression of exogenous transcripts[29]. Therefore, we fused the 3′ UTR of *nanos1* with ABE-Umax to create a germline-restricted base editor that we named ABE-Umax-nanos1 (Fig. 2c). We first targeted the *tubb4b* gene, which encodes proteins that form microtubules, essential components of the cellular skeleton. Mutations in TUBB4B have been linked to various neurological disorders that affect nervous system development and function[30]. A recent study identified a heterozygous missense variant (c.32 A > G, p.Q11R) in exon 1 of *TUBB4B*. This variant is a potential causative mutation in patients with early-onset hearing loss, hyperopia, hypophosphatemic rickets (HR), renal tubular Fanconi syndrome (FS), and nephrocalcinosis[31]. When developing the patient-relevant, precise disease model using ABE-Umax, we observed that the Q11R mutation in the fish *tubb4b* gene caused body curvature, hydrocephalus, and heart edema in all injected embryos (Fig. 2d, e). This mutation severely limited their survival to only 7 days. Conversely, embryos injected with ABE-Umax-nanos1 targeting the *tubb4b* site

exhibited normal development and reached adulthood for breeding (Fig. 2f). Genotyping of outcrossed F1 embryos from the breeding of the ABE-Umax-nanos1 F0 founder with AB wild-type fish demonstrated the tool's ability to introduce transmissible base edits in essential genes (Fig. 2g). Expanding this analysis, we targeted three additional essential genes (*hars1, ush1c, and eprs1*) using ABE-Umax-nanos1 editor and found that we could achieve up to 100% germline transmission rate of these biallelic edits to the F1 generation without affecting the viability of the F0 animals (Supplementary Fig. 5 and Supplementary Data 2). These data highlight the ability of ABE-Umax-nanos1 to efficiently edit the zebrafish germline, making it a valuable tool for generating biallelic modifications that would otherwise be early embryonic lethal.

Our results demonstrate that variants of ABE-Umax can generate targeted base changes in somatic cells and/or in the germline with high on-target editing efficiencies and low indel frequencies in zebrafish.

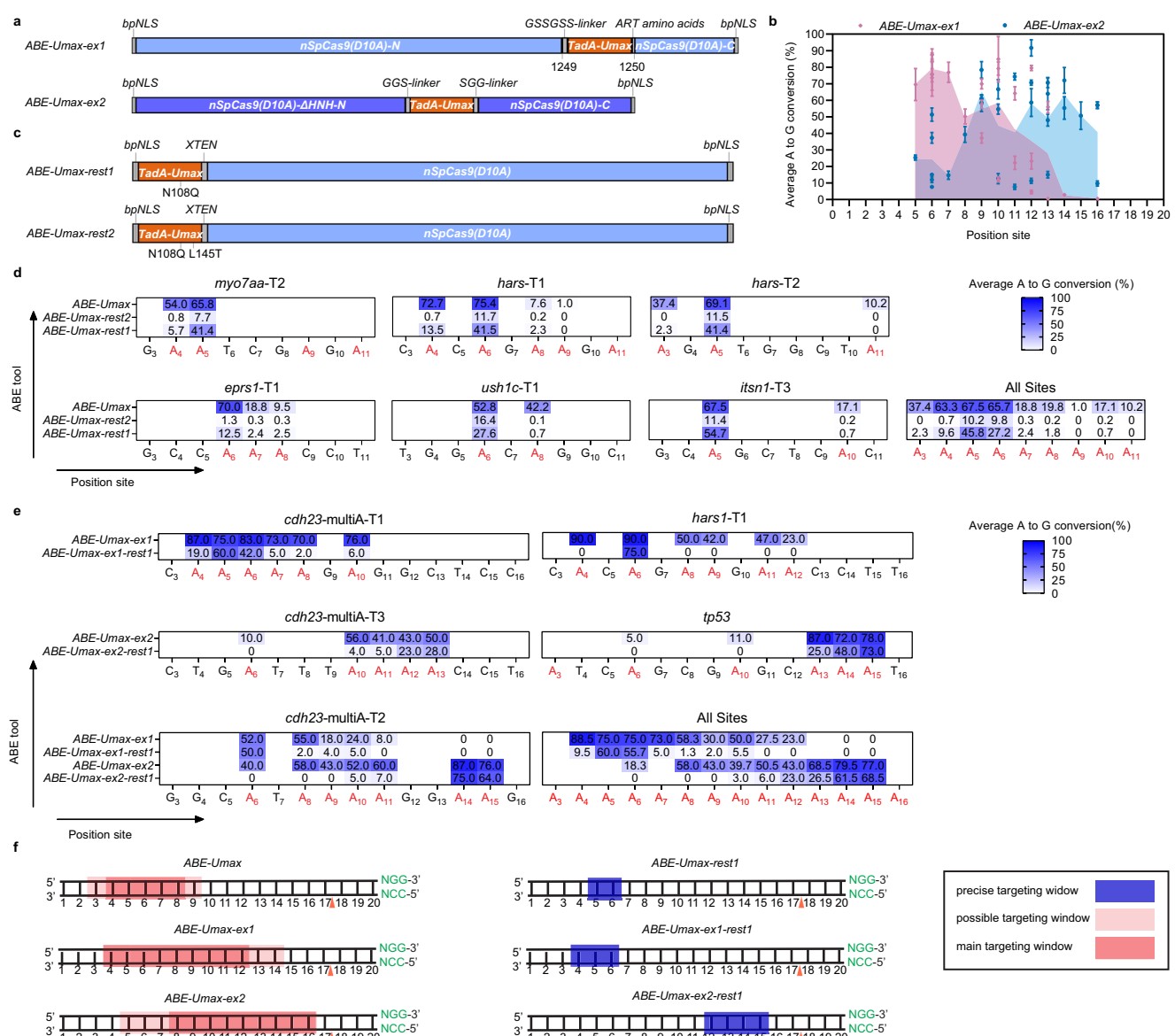

**Fig. 3 | Expanding editing windows for efficient and precise adenine base editing by ABE-Umax variants. a** Schematics showing constructs of ABE-Umax-ex1 and ABE-Umax-ex2 designed to shift the editing window of adenine base editing. **b** Assessment of the efficiency and targeting window of ABE-Umax-ex1 and ABE-Umax-ex2. Each data point represents the average editing activity at a particular site. The targeting window of ABE-Umax-ex1 is colored in pink from position 5 to 14, and the targeting window of ABE-Umax-ex2 is colored in blue from position 5 to 16, counting from the 5' terminal to the 3' terminal of the targeting site. Data from three independent experiments were analyzed. **c** The schematics of ABE-Umax-rest1 and ABE-Umax-rest2 constructs for precise adenine base editing. **d** Examining A-to-G editing efficiency of ABE-Umax, ABE-Umax-rest2, and ABE-Umax-rest1 across six endogenous genomic loci. Miseq was used to quantify A-to-G conversion frequencies at each adenine nucleotide within the 20-bp protospacer sequences. Results from three independent experiments were compiled, with editing

frequencies above 20% labeled in the heatmap. Color mapping (blue to white) represents editing efficiency from 100 to 0%. Editing base positions within the gRNA are indicated by numbers. **e** Evaluating the editing efficiency and targeting window of ABE-Umax-ex1-rest1 and ABE-Umax-ex2-rest1. EditR quantified A-to-G conversion frequencies at every adenine within the 20-bp protospacer. Data from three independent experiments were analyzed, with editing frequencies above 20% labeled in the heatmap. Color mapping (blue to white) indicates editing efficiency from 100% to 0%. Editing base positions within the gRNA are numerically labeled. **f** A schematic diagram illustrates the editing window for ABE-Umax tools. Editing preferences are indicated by a red shading gradient, with the darkest red marking the most preferred position and lighter shades representing less preferred edits. A blue line highlights the precise editing position. The red triangle denotes the theoretical SpCas9 cutting site, while the PAM sequence and its complement are shown in green. All source data in this figure are provided as a source data file.

## Expanding targeting coverage and editing windows of ABE editors

With the impressive efficiency of ABE-Umax, we next asked if we could create a suite of ABE-Umax effectors with altered editing windows to enable efficient targeting of a higher diversity of genomic locations. First, we attempted to broaden the editing window of ABE-Umax. We explored changing the location of the deaminase relative to the Cas9[13,14], which has shown some promise for widening the editing

windows of base editors in human cell culture but has not yet been tested in zebrafish. We generated two variants, ABE-Umax-extend1 (ABE-Umax-ex1) and ABE-Umax-extend2 (ABE-Umax-ex2) (Fig. 3a). For ABE-Umax-ex1, a TadA-Umax monomer was inserted into the docking site of SpCas9 at position 1249. This monomer has a GSSGSS linker at its N-terminus and an ART amino acid linker at its C-terminus, connecting the TadA monomer to Cas9 (Fig. 3a). For ABE-Umax-ex2, the HNH domain of Cas9 was deleted. Subsequently, a GGS-linker was used

to connect SpCas9 at position S793 to the N-terminus of TadA-Umax, and an SGG-linker was used to connect the C-terminus of TadA-Umax to SpCas9 at position R919 (Fig. 3a). By directing these variants toward 7 potential target sites in zebrafish, we found that ABE-Umax-ex1 mainly targets adenine bases located between positions 4 and 12 of the protospacer, while ABE-Umax-ex2 shifts the editing window closer to the PAM-proximal end of the protospacer (positions 8–16) (Fig. 3b) compared to typical ABEs that target positions 4–8 (Fig. 2b). Both effectors maintained high on-target efficiency (mean efficiency 51.71 and 44.93% for ABE-Umax-ex1 and ABE-Umax-ex2, respectively) and induced minimal indels, similar to ABE-Umax, as confirmed by sequencing (Supplementary Fig. 6).

Our data show that both extended window editors ABE-Umax-ex1 and ABE-Umax-ex2 can target previously inaccessible sites and provide greater flexibility in selecting efficient editors for any target site of interest. Importantly, both expanded-window tools and PAM-relaxed tools can enable targeting of sites that were previously inaccessible, and they can also significantly boost the efficiency at sites that were previously edited at very low efficiencies. For example, the editing efficiency of ABE-Umax-SpRY at position A5 on the protospacer of *rpl9-NCT* was only 5.33%, however, ABE-Umax-ex1 enabled 40% editing efficiency at the same position (Supplementary Fig. 7). Taken together, this work establishes a suite of ABE editors with expanded editing windows and/or PAM-less targeting that enable any adenine in the zebrafish genome to be targeted for A-to-G conversion.

### Improving editing precision by reducing bystander mutations

All editors developed here are efficient at editing adenine at the target site. However, they can also edit other neighboring adenines within the editing window, resulting in bystander mutations that would limit their application for both functional analysis of specific SNVs as well as therapeutic targeting. Recently, two ABE variants, ABE8e-N108Q and ABE9 (N108Q + L145T), were developed through structure-guided engineering to have a narrower editing window of 1–2 nucleotides at protospacer positions 5 or 6 (counting the PAM as positions 21–23)[32]. We aimed to test whether these same mutations could similarly narrow the editing window in our high-efficiency editor, ABE-Umax. We generated two variants which we named ABE-Umax-rest1 (harboring the N108Q mutation) and ABE-Umax-rest2 (carrying the N108Q/L145T double mutation) (Fig. 3c) and evaluated their editing efficiencies at six different sites in zebrafish. Both ABE-Umax-rest1 and ABE-Umax-rest2 exhibited precise base editing activity at protospacer positions 5 and 6, while virtually eliminating bystander editing at other adenines (Fig. 3d), consistent with the effect of these N108Q and N108Q/L145T mutations in human cells[32]. However, this came with reduced editing efficiency, with ABE-Umax-rest1 and ABE-Umax-rest2 achieving 54.67 and 15.01% of the unrestricted ABE-Umax efficiency at these sites, respectively (Supplementary Fig. 8). As ABE-Umax-rest2 exhibited an apparent reduction in on-target activity, we recommend ABE-Umax-rest1 for precise editing. To achieve narrower editing windows, we further introduced the N108Q mutation into ABE-Umax-ex1 and ABE-Umax-ex2 variants, which initially displayed broader PAM-proximal editing. These modified variants, named ABE-Umax-ex1-rest1 and ABE-Umax-ex2-rest1, demonstrated a remarkable ability to refine the editing window to target positions 4–6 and 12–15, respectively (Fig. 3e). Taken together, this work provides a toolkit of adenine base editors based on the highly efficient ABE-Umax that allow either very flexible or highly selective targeting of any desired genomic location (Fig. 3f, g).

### Off-target analysis of ABE-Umax and related base editors

We investigated off-target effects of TadA-Umax variants. Initially, we analyzed gRNA-dependent editing at the top three predicted off-target sites for *tyr*-g4 and *tp53* gRNAs (Fig. 4a and Supplementary Fig. 9a), which exhibited near-100% on-target efficiency (Fig. 2a, b). High-throughput sequencing revealed no significant DNA off-target

differences between ABE-Umax and ABE8e for either locus (Fig. 4b and Supplementary Fig. 9b). In fact, both tools showed slightly lower off-target effects (<10%) than the control group for the *tp53*-targeting gRNA (Supplementary Fig. 9b). Compared to ABE-Umax, the ABE-Umax-ex1 and ABE-Umax-ex2 variants displayed varying but acceptable (<10%) off-target effects for gRNAs targeting both *tyr*-g4 and *tp53* (Supplementary Figs. 10, 11). We also evaluated off-target effects for ABE-Umax-SpRY (a PAM-flexible Cas9 variant) using the same *tyr*-g4 and *tp53* gRNAs, finding no significant differences from ABE8e-SpRY or the control group (Supplementary Fig. 12). To examine potential RNA off-target effects, we performed transcriptome-wide RNA sequencing on zebrafish embryos injected with four constructs (ABE8e, ABE-Umax, ABE-Umax-ex1, ABE-Umax-ex2) with or without gRNAs targeting *tp53* or *tyr*-g4. We found that injecting the deaminase alone induced some RNA off-targets (in terms of edited adenines) (Fig. 4c and Supplementary Fig. 9c). However, ABE-Umax exhibited comparable RNA off-target activity to ABE8e for both gRNAs (Fig. 4c, d and Supplementary Fig. 9c, d). Notably, the ABE-Umax-ex1 shift-window tool revealed a slight increase in A-to-I RNA off-target edits for the *tyr*-g4 gRNA compared to ABE-Umax alone (Fig. 4c, d and Supplementary Fig. 9c, d). Overall, the ABE-Umax variant, did not induce significant off-target effects at both DNA and RNA levels compared to the ABE8e variant. However, shift-window variants of both editors can induce varying off-target effects at both DNA and RNA levels, likely due to changes in Cas9 and deaminase positioning.

### Analysis of candidate genetic variants in human disease genes by inducing biallelic mutations

One of the most powerful features of CRISPR-Cas9 genome editing is the ability to induce biallelic mutations, thereby creating homozygous loss-of-function animals without the cumbersome need for multi-generational crosses. In zebrafish, this means that one can evaluate gene function rapidly and efficiently in the F0 (founding) generation[33], which has been explored at a large scale using CRISPR-Cas9 gene knockouts[33–35] but, to our knowledge, not using more precise tools like base editors to install SNVs. Given the high efficiency of ABE-Umax and its variants, we investigated whether these modified editors can be used to induce biallelic mutations to evaluate the pathogenicity of human missense mutations, including variants that are predicted to be benign or variants of unknown significance (VUS), in the zebrafish model (Fig. 5a, b).

Around 80% of prelingual hearing loss cases are linked to genetic factors, with ~70% of these cases falling under non-syndromic hearing loss, involving over 100 genes[36]. The zebrafish lateral line system is a useful model for investigating hearing loss because it shares similarities with mammalian inner ear hair cells, and we can conveniently assess hair cell development and function by employing live dyes such as Yo-Pro-1 (Fig. 5c). We focused on selecting orthologs of three pivotal genes associated with human hearing loss, namely *MYO7A*, *USH1C*, and *CDH23*. Mutations in these genes can lead to Usher syndrome, a genetic condition causing combined deafness and blindness, as well as non-syndromic hearing loss.

Previous studies have demonstrated that zebrafish with loss-of-function mutations in *myo7aa* or *ush1c* exhibit dysfunctional hair cells in both the inner ear and lateral line. This dysfunction results in a measurable loss of the typical startle response triggered by sharp vibrations in their aquatic environment[37,38]. The human missense mutations in *MYO7A* (c. 1208 A> G, Y403C), and *USH1C* (c. 1 A >G, M1V) are predicted as likely pathogenic in the ClinVar database (Supplementary Fig. 13), but the pathogenicity of these mutations has never been validated. To functionally test these variants, we designed one guide RNA for each orthologous site in zebrafish (Fig. 5d, e) and injected zebrafish embryos with ABE-Umax mRNA and guide RNA. Strikingly, we found that the mean base conversion rate (three pools, each pool containing eight randomly selected embryos) was 95% at the

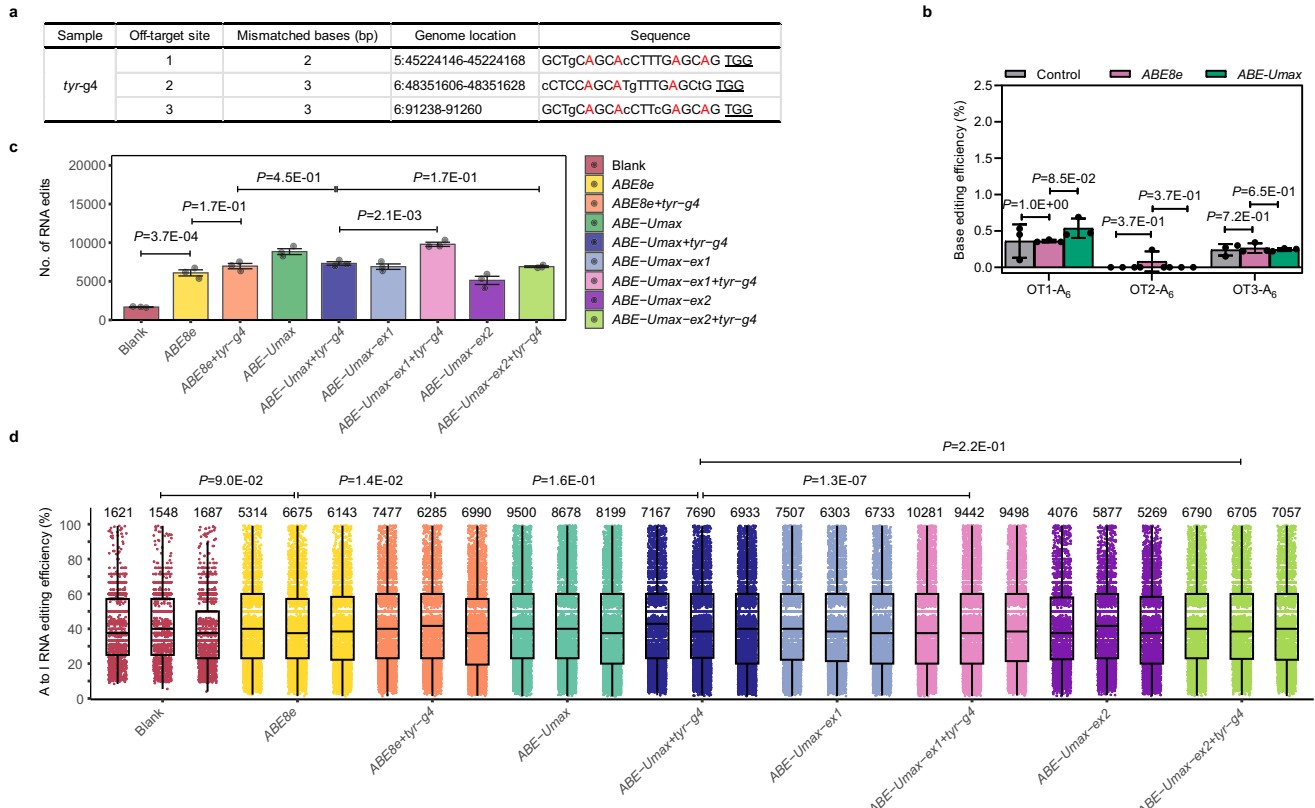

**Fig. 4 | Off-target analysis of ABE-Umax-related tools in zebrafish. a** Potential off-target sites at the *tyr*-g4 locus. The top three high-scoring off-target sites (PAM are underlined) are shown. Mismatched bases are indicated in lowercase, and the potentially editable A is highlighted in red. **b** DNA off-target comparison of ABE78e and ABE-Umax at the *tyr*-g4 locus. Editing efficiencies are displayed with error bars indicating mean ± s.d. (*n* = 3 biological replicates). Two-tailed paired *t*-test were performed (with *P* values marked). **c** Transcriptome analysis of edited adenine nucleotides in zebrafish embryos. Embryos were injected with ABE8e+*tyr*-g4, ABE-Umax-ex1+*tyr*-g4, ABE-UMax-ex2+*tyr*-g4, or their related mRNA only. Data from three independent replicates are shown. Two-tailed paired *t*-test were performed (with *P* values marked). **d** RNA A-to-I conversion frequencies in injected zebrafish embryos. Representative jitter plots display frequencies for embryos injected with ABE8e+*tyr*-g4, ABE-Umax-ex1+*tyr*-g4, ABE-UMax-ex2+*tyr*-g4, or their related mRNA only. The numbers of A-to-I RNA edits are indicated upon the plots. Data from three independent replicates are shown. All box plots include the median line, the box denotes the interquartile range (IQR), whiskers denote the rest of the data distribution and outliers are denoted by points greater than ±1.5 × IQR. Two-tailed paired *t*-test were performed (with *P* values marked). All source data in this figure are provided as a source data file.

*myo7aa* allele (Fig. 5f), and 70% at the *ush1c* locus (Fig. 5g). While control animals exhibited normal Yo-Pro-1 uptake by hair cells of lateral line and inner ear, both base-edited variants failed to uptake the dye. This indicates a disruption of hair cell function, which we attribute to the loss of function in either *myo7aa* or *ushc1c* genes (Fig. 5f, g). Overall, our results demonstrated that ABE-Umax can be used to rapidly evaluate the function of human pathogenic SNVs in F0 zebrafish. Importantly, our work functionally validated the causal pathogenic impact of *myo7aa* (Y414C) and *ushc1c* (M1V), which was until now only suspected based on correlation. Building on this success, we directed ABE-Umax to install SNVs in four genes implicated in neurodevelopmental disorders (*eprs1*, *spata5l1*, *cog1*, and *adam22*), and achieved 100%; 100%; 95%; and 90% editing, respectively (Supplementary Fig. 14).

Extending our study beyond exonic SNVs, we next focused on splicing mutations, which are estimated to be responsible for about 10% of human genetic diseases[39] and are among the most prevalent classes of human pathogenic mutations. ABEs would be capable of targeting both the 5′ splice donor (GT) and the 3′ splice acceptor (AG) sites (Fig. 6a), allowing for control over exon skipping, alternative splicing, or generating functional gene knockouts (Fig. 6b)[40]. Here, we used ABE-Umax editors to introduce two types of splicing mutations in zebrafish. First, we targeted the splice donor site of intron 2 within the *cdh23* gene (Fig. 6c), which is associated with non-syndromic autosomal recessive deafness. In this case, the expected A mutation was at

position 12 from the 5′ end, so we use ABE-Umax-ex2 (Fig. 6c). Second, we targeted the splice acceptor site of intron 3 in the *tyr* gene (Fig. 6d), which is associated with albinism. The expected A mutation was at position 6 from the 5′ end, so we could use ABE-Umax. After injecting zebrafish embryos with the relevant effector-expressing mRNA and guide RNAs, we found that 100% of the embryos (8/8) had the desired A to G edits at *cdh23* and *tyr* loci (Fig. 6e, f). Consistent with the expected phenotypic effects of these splice site mutations, we observed loss of Yo-Pro1 positive functional hair cells in *cdh23* mutant embryos (Fig. 6e) and a near-complete loss of pigmentation in the *tyr* mutant embryos (Fig. 6f). We confirmed that ABE-Umax editing induced cryptic pre-mRNA splicing defects at mRNA level in both loci (Fig. 6g, h). These results demonstrate the potential of ABE-Umax editors as a robust toolset for generating splicing-induced disease models in zebrafish.

We next investigated the effectiveness of these editors in targeting patient-specific pathogenic variants. Waardenburg syndrome is a genetic disorder causing hearing loss and pigmentation changes in hair and skin. Mutations in the *MITF* gene are responsible for about 20% of human cases. A specific point mutation (S357P, corresponding to zebrafish S245P) in *MITF* has been suggested as a potential disease-causing mutation in patients[41,42]. To create a patient-specific disease model, we began by analyzing the zebrafish *mitfa* locus. We identified two potential gRNA target sites: one containing an NGC PAM sequence (gRNA1, shown in blue) and another containing an NGG PAM (gRNA2,

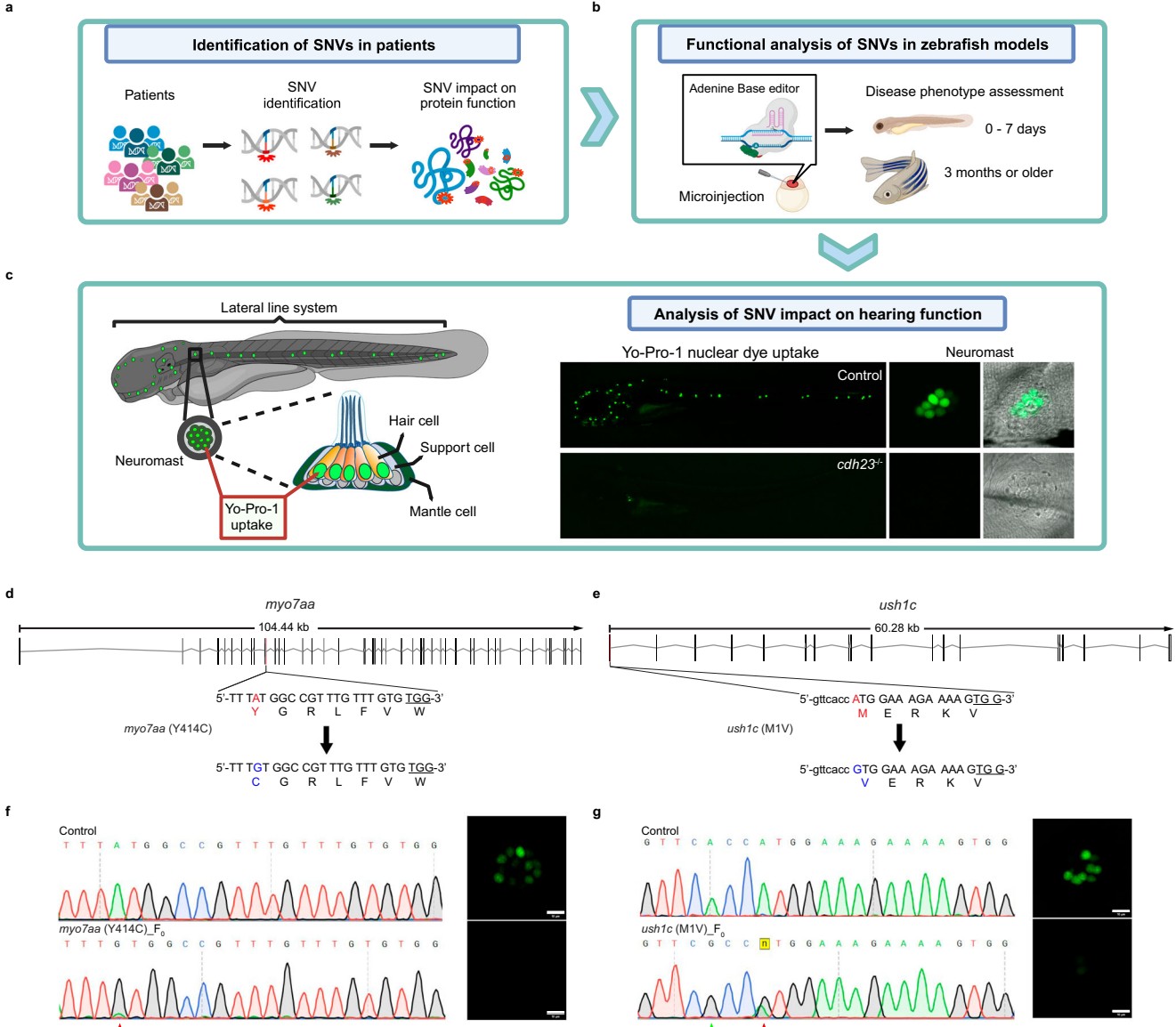

**Fig. 5 | Functional analysis of VUS sites in F0 generation using ABE-Umax and its variants. a**, **b** Overview of the variant functional analysis pipeline designed to test variants with unknown functions in the F0 generation at both larval and adult stages, using various morphological, cellular, and behavioral phenotypes. Created with BioRender.com released under a Creative Commons Attribution-NonCommercial-NoDerivs 4.0 International license. **c** Schematic of the zebrafish lateral line sensory system, which comprises clusters of mechanosensory hair cells known as neuromasts. Lateral line hair cells are functionally and molecularly similar to the hair cells found in the inner ear of zebrafish and serve as an excellent model for studying hearing function (left panel). A whole-mount live dye stain of hair cells using Yo-Pro1 dye, which is taken up by hair cell mechanotransduction channels. Control embryos display a green cluster of hair cells, whereas $cdh23^{-/-}$ mutants

lack Yo-Pro1 positive hair cells, indicating the absence of functional hair cells. Created with BioRender.com released under a Creative Commons Attribution-NonCommercial-NoDerivs 4.0 International license. **d**, **e** Schematic diagrams of *myo7aa* (Y414C) and *ush1c* (M1V) in zebrafish. The targeted sequence is displayed with the PAM underlined. The targeted nucleotide and amino acid are highlighted in red, while the expected changes in nucleotide and amino acid are highlighted in blue. **f**, **g** Sequencing results and phenotypes of *myo7aa* (Y414C) F0 (**e**) and *ush1c* (M1V) F0 (**f**) induced by ABE-Umax. The red arrowhead points to the expected nucleotide substitutions, while a green arrowhead in the Sanger sequencing chromatograms indicates bystander base substitutions. Neuromast hair cells labeled with Yo-Pro1 (green) are displayed adjacent to the chromatograms. Three independent experiments were repeated with similar results. bars: 10 μm.

shown in green) (Fig. 7a). Our initial experiment involved injecting ABE-Umax-SpRY mRNA and *mitfa* gRNA1 into zebrafish embryos to assess adenine base conversion. Although we detected overlapping C/T peaks at the targeted cytosine (position 6 from the 5′ end) in pools of eight randomly selected embryos, the desired edit efficiency was only 8% (Fig. 7b). Switching to the ABE-Umax-ex2 tool with *mitfa* gRNA2 targeting the same site resulted in higher efficiency of 43%, but we observed significant unintended edits at other locations within the large editing window (Fig. 7c). In contrast, the restricted editor, ABE-Umax2-rest1, achieved precise and efficient single-base edits at the desired site with a 40% efficiency, and without any apparent bystander

editing (Fig. 7c). Genotyped F2 embryos displayed pigment deficiency phenotypes at 2 days post-fertilization (dpf) (Fig. 7d, e), underlining the importance of the S357 residue for MITF function. In conclusion, these data demonstrate the high value, versatility, efficiency, and expanded editing potential of the ABE-Umax editors.

**Strategy for selecting the appropriate adenine base editor (ABE)**
We have presented several ABE-Umax editors with diverse characteristics. When evaluating a specific ABE, selecting the appropriate tool becomes a fundamental issue. Our strategy is as follows (Fig. 7f): Initially, we examine the presence of a GG motif within 6–20 base pairs

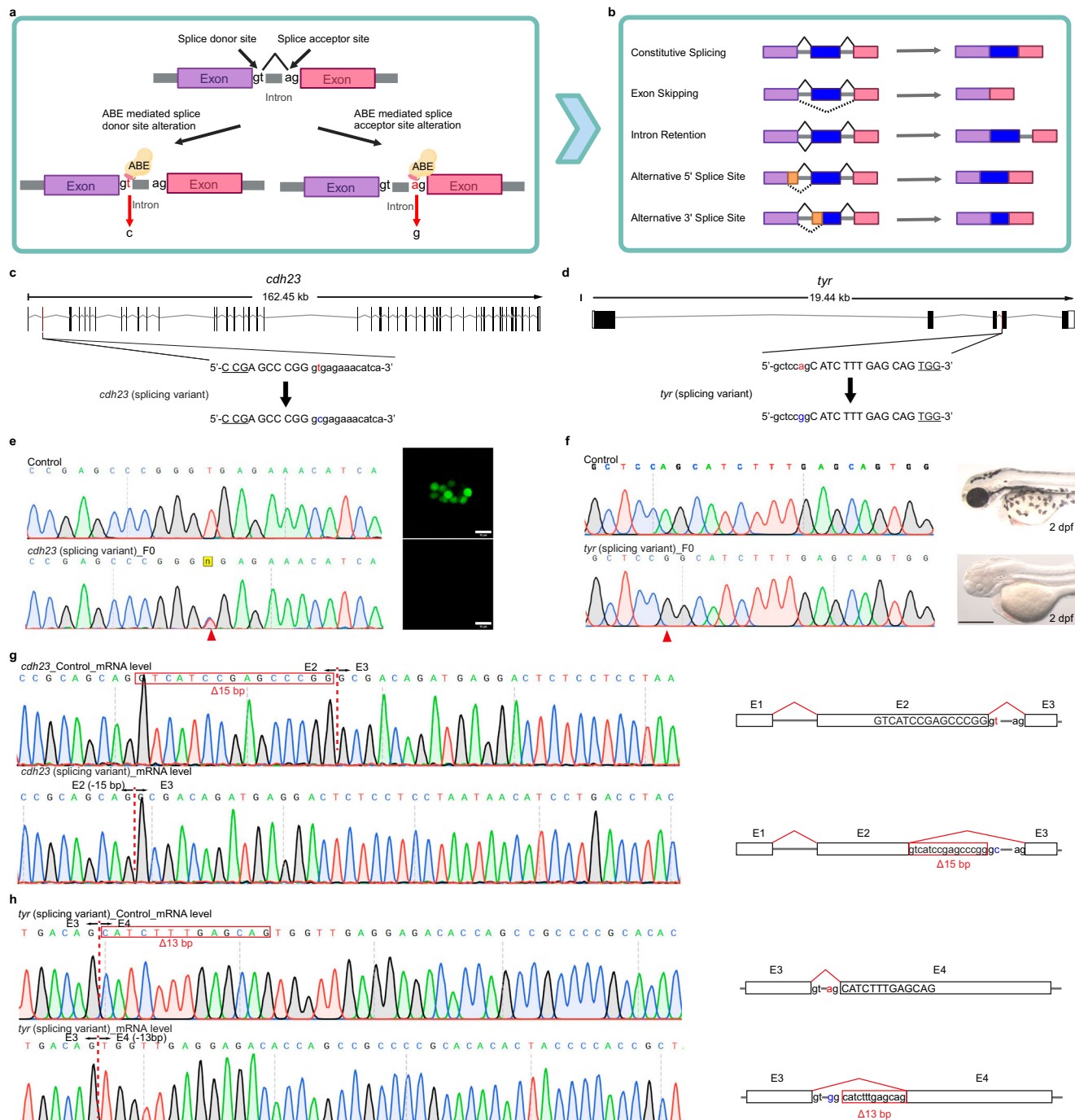

**Fig. 6 | Editing of splice variants using ABE editors. a** A schematic diagram of adenine base editing tools inducing splicing mutations. The colored bars (purple and pink) represent exons. Created with BioRender.com released under a Creative Commons Attribution-NonCommercial-NoDerivs 4.0 International license. **b** Overview of possible common alternative splicing events. The mechanisms of the most common alternative splicing events, which include Exon Skipping, Intron Retention, Alternative 5′ splice site selection, and Alternative 3′ Splice Site selection, are shown. The colored bars (purple, blue, and pink) represent exons. Yellow bars represent the alternatively spliced exons, while the gray lines connecting them represent intronic segments. Created with BioRender.com released under a Creative Commons Attribution-NonCommercial-NoDerivs 4.0 International license. **c**, **d** Schematic diagrams of the *cdh23* (**c**) and *tyr* (**d**) splicing loci. The targeted sequence is displayed with the PAM underlined. The targeted nucleotides are highlighted in red, while the expected nucleotide changes are highlighted in blue.

**e** Sequencing results and phenotype of the *cdh23* splicing variant induced by ABE-Umax-ex2. The red arrowhead indicates the expected nucleotide substitutions. Neuromast hair cells labeled with Yo-Pro1 (green) are displayed adjacent to the chromatograms. Three independent experiments were repeated with similar results. Scale bars: 10 μm. **f** Sequencing results and phenotype of the *tyr* splicing variant induced by ABE-Umax. The red arrowhead indicates the expected nucleotide substitutions. Three independent experiments were repeated with similar results. Scale bars: 500 μm. **g** Sanger sequencing of cDNA demonstrates *cdh23* splicing defects caused by ABE-Umax-ex2. The T-to-C substitution at the Exon 2 splicing donor site in *cdh23* induces a 15 bp deletion at the mRNA level. The deletion sequence is highlighted in a red box. **h** Sanger sequencing reveals *tyr* splicing defects induced by ABE-Umax. The A-to-G substitution at the Exon 3 splicing acceptor site in *tyr* results in a 13 bp deletion at the mRNA level. The deletion sequence is highlighted in a red box.

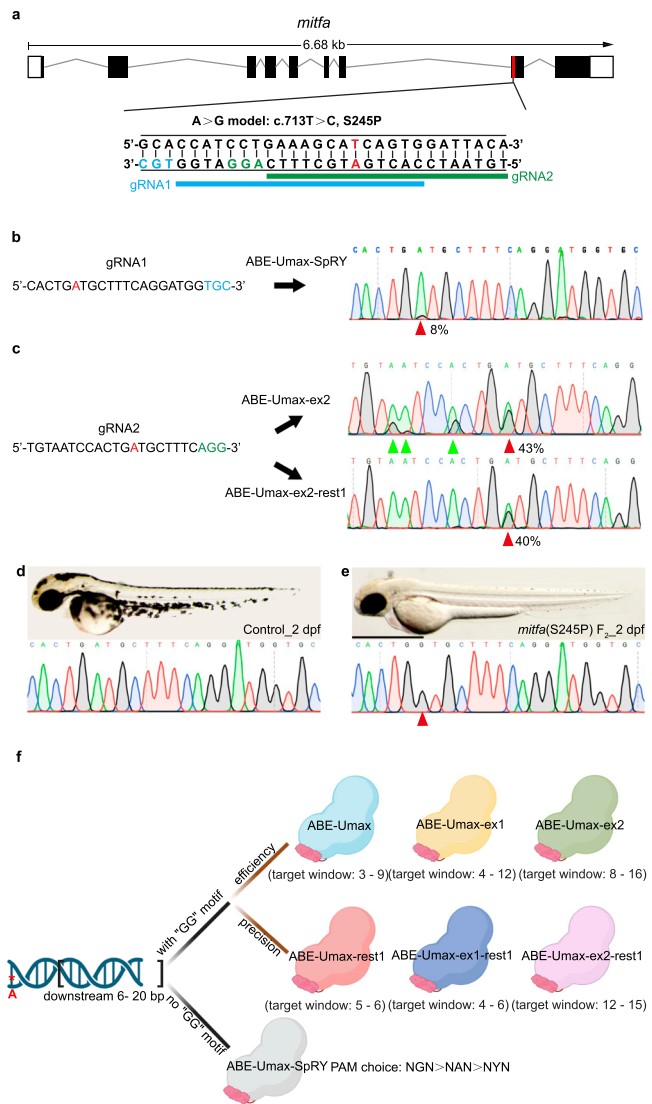

**Fig. 7 | Disease modeling using ABE-Umax editors. a** Generating A-to-G conversion in a disease-relevant zebrafish model (*mitfa* c.713 T > C). Protospacers are shown in black with PAM sequences in blue or green. The desired adenine for editing is highlighted in red. Potential gRNA target sites are marked with blue and green lines, respectively. **b** Sequencing results for *mitfa* (S245P) in ABE-Umax-SpRY-induced F0 generation. The desired adenine for editing is highlighted in red. Protospacers are shown in black with PAM sequences in blue. The red arrowhead indicates the expected nucleotide substitution. **c** Sequencing results for *mitfa* (S245P) in F0 generation induced by ABE-Umax-ex2 and ABE-Umax-ex2-rest1. The desired adenine for editing is highlighted in red. Protospacers are shown in black with PAM sequences in green. The red arrowhead indicates the expected nucleotide substitution. Bystander base substitutions are marked with green arrowheads in the Sanger sequencing chromatograms. **d** Lateral view of wild-type embryos at 2 dpf with sequencing results. Scale bar: 1 mm. **e** Lateral view of F2 homozygous embryos with the *mitfa* (S245P) mutation at 2 dpf, exhibiting pigmentation defects. The red arrowhead indicates the expected nucleotide substitution. Scale bar: 1 mm. **f** Selection of the right tool for A-to-G editing depends on the presence of a specific DNA sequence motif and can be selected as follows: Look for a GG motif within 6–20 base pairs downstream of the target adenine (red asterisks). Option 1: If a GG motif is present, (1) for high efficiency, consider ABE-Umax (window 3–9), ABE-Umax-ex1 (window 4–12), or ABE-Umax-ex2 (window 8–16); (2) for high precision, choose ABE-Umax-rest1 (window 5–6), ABE-Umax-ex1-rest1 (window 4–6), or ABE-Umax-ex2-rest1 (window 12–15). (3) If multiple tools meet your criteria, try them all to see which one works best. Option 2: If no GG motif is found, (1) use ABE-Umax-SpRY; (2) prioritize PAM sequences in this order: NGN > NAN > NYN; (3) position the edited adenine within the 4th–8th position of the editing window. The schematic diagrams in different colors represent different adenine-based editing tools. Created with BioRender.com released under a Creative Commons Attribution-NonCommercial-NoDerivs 4.0 International license.

downstream of the hypothetical target adenine for editing. If such a motif is identified, we consider a series of tools recognizing NGG PAM sequences. Prioritizing efficiency leads to the selection of ABE-Umax (window = 3–9), ABE-Umax-ex1 (window = 4–12), and ABE-Umax-ex2 (window = 8–16), while emphasizing precision prompts the selection of ABE-Umax-rest1 (window = 5–6), ABE-Umax-ex1-rest1 (window = 4–6), and ABE-Umax-ex2-rest1 (window = 12–15). If multiple tools meet the criteria, it is advisable to experiment with each to determine the one demonstrating optimal performance. If no GG motif is found, then ABE-Umax-SpRY should be selected. When using this tool, precedence should be given to PAM preference, with NGN > NAN > NYN, ensuring the edited adenine is positioned within the 4th–8th position (on the protospacer) of the window. Overall, the suite of ABE-Umax effectors we developed here has great potential to overcome many of the limitations of previous base editors for disease model generation and potential gene therapy.

## Discussion

While base editors can alter a single DNA base within the genome without causing double-stranded DNA breaks, their utility is hampered by various factors, including efficiency, a limited targeting window, PAM restrictions, editing precision, and the potential for off-target effects[43]. The quest for a perfect base editor remains a challenge. Several recent studies have created base editor variants with PAM-less Cas9, orthologous Cas9, engineered TadA

enzyme. However, many of these tools are either not tested extensively in vivo or suffer from several limitations such limited targeted coverage and editing window and low efficiencies[11,44–46]. In this study, we overcame many of these previous limitations to successfully establish a set of highly efficient ABEs based on ABE-Umax, a modified base editor with robust in vivo editing efficiency and low indel frequency in zebrafish, which can likely be easily applied in other model organisms such as *C. elegans*, *Drosophila*, *Xenopus*, mice, and other fish species.

An inherent challenge in using base editors in zebrafish and many other model organisms is their low efficiency and strong locus bias[20]. Here, we pressure-tested ABE-Umax and its derivatives by targeting 70 genomic loci across 36 genes—including coding and non-coding sites of interest in disease-related genes located on different chromosomes—and found that we could achieve close to 100% biallelic editing efficiency on many tested targets. This improvement over the current state-of-the-art base editors allows for the induction of biallelic missense mutations in the F0 injected embryos: an indispensable capability for high-throughput screening of SNV phenotypes.

Our extensive optimization efforts led to developing several modified editors with expanded or shifted editing windows, flexible PAM recognition, and enhanced editing precision. These base editors offer a range of options for editing any adenine of interest, enabling the tailored selection of the most suitable editor for each target site, as we demonstrated in our targeting of the *mitfa* pathogenic mutation implicated in Waardenburg syndrome. When choosing an appropriate editor for a given target site, our data suggests that NGG PAM sites often lead to the most efficient editing, pointing to the ABE-Umax effector's superiority for these sites if efficiency is the primary consideration. However, in some cases, the target adenine may fall outside the position 4–8 editing window, in which case ABE-Umax-ex1 (editing window 3–12) or ABE-Umax-ex2 (editing window 8–16) can enable efficient targeting with an NGG PAM. In cases where there are no suitable NGG PAMs within the vicinity of the target adenine, the PAM-less tool ABE-Umax-SpRY provides flexibility to target these sites with high

efficiency, albeit potentially at the expense of higher off-target editing. Importantly, even with specified PAM preferences, editing efficiency can vary. When using ABE-Umax-SpRY, it is recommended to test multiple guide RNAs (gRNAs) to identify the one with the best performance. Finally, in cases where bystander editing of non-target adenines within the editing window must be avoided, we provide ABE-Umax-rest1, ABE-Umax-ex1-rest1, and ABE-Umax-rest2 with tighter editing windows (5–6, 4–6, and 12–15, respectively) to achieve higher precision for only the target adenine, despite a generally lower editing efficiency. Taken together, the ABE-Umax toolkit we present here provides many options to target adenines of interest across the genome and allows the user to choose the appropriate tool for each site, dependent on the relative importance of efficiency and fidelity for the application at hand.

Another significant challenge in base editing is the occurrence of off-target effects, which pose a considerable concern in clinical applications. While ABE-Umax demonstrates robust editing activity, we observed no significant off-target effects compared to ABE8e at either the DNA or RNA level (except for shift-window tools, which exhibited some off-target effects). This is likely due to changes in the relative positioning of the deaminase and Cas9. However, in zebrafish, off-target mutations are not a significant concern since they can be effectively mitigated by back-crossing.

Alternatively, Cas9 moiety can be replaced by high-fidelity Cas9 variants that have been shown to have significantly reduced off-targets[47–49]. There is often a trade-off between overall activity and specificity in high-fidelity variants. Recently, a Cas9 variant known as Sniper2L was reported to exhibit high fidelity while maintaining a high activity level[50].

Altogether, the data presented in this study demonstrate that the ABE-Umax family of base editors can efficiently and precisely access and edit single-nucleotide variants in zebrafish, expanding the potential targeting space beyond what was possible with existing ABEs. We use these effectors to demonstrate the causality of variants of unknown significance, investigate coding and non-coding mutations in disease-associated genes, and create biallelic mutations in essential genes through germline-restricted editing.

## Methods
### Ethical statement
All zebrafish experiments were carried out as per protocol 20-07 approved by the Institutional Animal Care Committee (IACUC) of OMRF and by the University Animal Care and Use Committee of South China Normal University (SCNU-SLS-2021-008).

### Zebrafish maintenance
Wild-type zebrafish strain NHGRI-1[51] and *mitfa*<sup>W2/W2</sup> mutants were raised and maintained at 28.5 °C on a 14 h light/10 h dark cycle. The selection of mating pairs (12–15 months) was random from a pool of 30 males and 30 females. Except for the experiments screening high-efficiency base editing tools in Fig. 1b, c, all other experiments were conducted using wild-type zebrafish.

### Plasmid construction
All ten TadA variant plasmids were constructed based on plasmid zSpRY-ABE8e (35701400), incorporating their respective mutations. For plasmids ABE-Umax-rest1 and ABE-Umax-rest2, one mutation (N108Q) and two mutations (N108Q + L145T) were introduced into ABE-Umax, respectively. In the case of plasmid ABE-Umax-ex1, the TadA-Umax monomer was inserted into the docking site of SpCas9 at position 1249, chemically synthesized based on pT3TS-zSpCas9. A GSSGSS linker and an ART amino acids linker were employed to bridge the gap between the N-terminus and C-terminus of the TadA-Umax monomer and Cas9, respectively. For plasmid ABE-Umax-ex2, the HNH domain of Cas9 was deleted, also using pT3TS-zSpCas9 as a template.

Then, a GGS-linker was utilized to connect SpCas9 S793 to the TadA-Umax N-terminus, and an SGG-linker was used to join the TadA-Umax C-terminus to SpCas9 R919. In constructing the ABE-Umax-nanos1 construct, the nanos1 3′ UTR and SV40 late polyA signal sequences was chemically synthesized. All fragment fusions and mutations were generated using the Vazyme Mut Express II Fast Mutagenesis Kit V2 (Cellagen Technology LLC, CA, USA) with primers listed in Supplementary Data 3. The assembled plasmids were then transformed and amplified in DH5α Chemically Competent Cells (New England Biolabs, MA, USA).

### Single guide RNAs (sgRNAs) and mRNA synthesis
All of the sgRNAs in this study were synthesized with chemical modifications, including MS (2′-O-methyl (M), 2′-O-methyl 3′phosphorothioate) modifications at both ends, by GenScript Inc. (NJ, USA) and Synthego Inc. (CA, USA). The target sequences are listed in Supplementary Data 4. All mRNAs, except for ABE-Umax-nanos1, were transcribed in vitro from a template linearized with *XbaI* (NEB, USA) using the T3 mMESSAGE mMACHINE Kit (Thermo Fisher Inc., CA, USA) and purified with the Monarch RNA Cleanup kit (New England Biolabs, MA, USA). For ABE-Umax-nanos1, the template was linearized with *NotI*, and the SP6 mMESSAGE mMACHINE Kit (Thermo Fisher Inc., CA, USA) was used. The capped mRNAs and synthetic sgRNAs were dissolved in a 2000 ng/μl stock solution and stored at −80 °C.

### Microinjection, morphological phenotyping, and imaging
A mixture containing 200 ng/μl of sgRNA and 400 ng/μl of Cas9 mRNA in 2 nl was injected into one-cell-stage embryos. After 2–5 days post-fertilization (dpf), embryos were anesthetized with 0.016% tricaine/MS-222 (Sigma-Aldrich, MO, USA) and manually oriented in 3% methylcellulose (Sigma-Aldrich, MO, USA) for imaging. The number of melanocytes rescued in *mitfa*<sup>W2/W2</sup> mutant embryos at 2 dpf was used to assess the efficiency of ABE variants. To visualize neuromast hair cells, embryos at 5 dpf were incubated in 1 mM YO-PRO™−1 (Invitrogen, USA) for 1 h at room temperature. All images were captured using an Olympus SZX12 stereomicroscope with an Olympus DP71 color digital camera (Olympus, Japan). After imaging, embryos were genotyped and traced to establish the phenotype-genotype relationship.

### Variant pathogenicity prediction
For predicting the pathogenicity of the variants, we used an on-line prediction tool, VARITY[52].

### mRNA splicing analysis
For splicing analysis, total RNA was extracted from embryos with a phenotype using TRIzol Reagent (Invitrogen, USA), and the Superscript-III One-Step kit (Invitrogen, USA) was used to get the targeted fragments following the manufacturer's instructions. Then the fragments were cloned into the pCR4-TOPO vector (Invitrogen, USA) for Sanger sequencing. Splicing variants were aligned with original transcripts using CLC Sequence Viewer 8.0 (Qiagen, Germany) and then analyzed manually.

### Base editing analysis and genotyping
For base editing analysis, genomic DNA was extracted from three pools, each containing six randomly collected. The targeted locus 150–300 bp in length was amplified using HotStart Taq-Plus DNA polymerase (Qiagen, USA) with the primers in Supplementary Data 4 and purified with kit DNA Clean Concentrator-5 (ZYMO, USA) for Sanger sequencing. The direct Sanger sequencing results were analyzed by EditR (1.0.10) program[19,24]. For genotyping, the genome was extracted from every single embryo after imaging or functional analysis.

## Screening for germline transmission events

To identify embryos with successful germline transmission of the edited locus, we collected two groups (approximately five embryos per group) of sibling embryos at 2 days post-fertilization (2 dpf) from F1 generation fish. Genomic DNA was extracted from the collected embryos, and the targeted locus was examined for base editing events using PCR and Sanger sequencing. The germline targeting efficiency was calculated as the percentage of positive founders (showing base editing) among all screened founders. For each positive founder fish, we evaluated the transmission rate by analyzing approximately 8 embryos from their F1 progeny individually using PCR and Sanger sequencing.

## Next-generation sequencing (NGS) and analysis

Genomic DNA was extracted from wild-type or injected embryos by alkaline lysis buffer-based DNA extraction. After PCR fragments purification, the samples were subjected to two paired-end read sequencing using the Miseq strategy (Illumina) and the sequencing data were analyzed using CRISPResso2[53].

## DNA Off-target analysis

For each gRNA, we predicted potential off-target sites using Cas-OFFinder[54] and CRISPOR (Version 4.99)[55]. We then used CRISPOR (Version 4.99) to calculate specificity scores for each site. Finally, the top three highest-scoring off-target sites were selected for further evaluation of off-target effects using next-generation sequencing (NGS).

## RNA-seq experiments and data analysis

Two independent batches of samples were collected from each experimental group for RNA-seq analysis. Whole embryos (40 per group) at 2 days post-fertilization (dpf) were used for total RNA extraction using TRIzol® Reagent (Life Technologies, USA) according to standard protocols. RNA quality and concentration were assessed with a Nanodrop2000 spectrophotometer (Thermo Fisher Scientific, USA). We ensured total RNA concentrations were at least 200 ng/μl with a minimum RNA integrity number (RIN) of 7 and a 28S/18S ratio of 1.0 or greater. Oligo (dT) beads were used to enrich mRNA, followed by fragmentation (200–700 nt) and reverse transcription into first-strand cDNA with random primers. The second-strand cDNA underwent end repair, polyA addition, and adapter ligation for Illumina sequencing. After purification with a Quick PCR extraction kit, the products were amplified by PCR and sequenced using the Illumina HiSeqTM platform. Established procedures were followed for RNA-seq data analysis[56–58]. Clean data were obtained from raw FASTQ data by removing adapter sequences and trimming low-quality bases using fastp (version 0.18.0). HISAT2.2.4 was used to align paired-end reads from each replicate to the GRCz11 zebrafish reference genome (UCSC). Variant calling was performed using the GATK best practices pipeline with GATK 3.8. Single-nucleotide variants (SNVs) with more than ten reads were filtered and categorized as A-to-G substitutions to evaluate RNA off-target activities. Finally, the number of adenosines converted to inosines in each sample's transcriptome was calculated. REDItools v1.3 was used to quantify A-to-I editing in each sample.

## Statistics and reproducibility

No statistical method was used to predetermine the sample size. No data were excluded from the analyses and samples were randomized. Each experiment was repeated at least three times. Every sample size is described in the figure legends or the Source Data. Data were displayed as the mean value ± standard deviation (SD). Statistical analysis was performed using GraphPad Prism version 8.0.2 (GraphPad Software, USA). The significance level was set to <0.05 (*), <0.01 (**), and <0.001 (***). The $P$ value was described as follows. Two-tailed unpaired Student's $t$-test was used to compare base editing efficiency between different groups. Two-tailed paired Student's $t$-test of nonparametric Wilcoxon matched-pairs signed rank test was used for mean editing efficiency comparisons between two groups.

## Reporting summary

Further information on research design is available in the Nature Portfolio Reporting Summary linked to this article.

## Data availability

NGS data were available on the National Center for Biotechnology Information Sequencing Read Archive (SRA) database under project numbers PRJNA1118438[59] and PRJNA1118794[60]. Plasmids from this study are available at Addgene under plasmid numbers #222139–#222144 (https://www.addgene.org/222139/, https://www.addgene.org/222138/, https://www.addgene.org/222140, https://www.addgene.org/222141/, https://www.addgene.org/222142, https://www.addgene.org/222143, and https://www.addgene.org/222144). All data supporting the findings of this study are available within the article and Supplementary Information files. Source data are provided with this paper.

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

## Acknowledgements

This work was partly supported by US National Institutes of Health (NIH) grant R01GM136741 (G.K.V. and P.M.), and Presbyterian Health Foundation (G.K.V.). Y.L. and F.L. are supported by the National Natural Science Foundation of China (32070819 to Y.L. and 32300692 to F.L.). P.M. is supported by R01 HL151576, P50 HD109861, and the Simons Foundation. We thank Dr. April Pawluk from Life Science Editors for her help with editing this manuscript.

## Author contributions

W.Q. and F.L. performed the experimental work and analyzed the results. S.-J.L., C.P., K.H., Y.Z., L.L., and P.V. contributed to the experiments. W.Q., F. L., and G.K.V. conceived the project. W.Q., F.L., P.M., and G.K.V. wrote the original draft. All other authors reviewed and edited the manuscript. G.K.V. acquired funding and supervised the study.

## Competing interests

The authors declare no competing interests.
