## [Peer Review File · Nature Communications]

REVIEWER COMMENTS

Reviewer #1 (Remarks to the Author):

The authors utilized albino/nacre *mitfa*^{W2/W2} zebrafish to optimize ABE. They showed that, within the ABE-Umax platform, a shifted, narrowed, or broadened editing windows, reduced bystander mutation frequency, as well as highly flexible PAM sequence requirements could be achieved. The study is well designed, and the new base editing tool may be of broad use for zebrafish genetic engineering. While, its advantage over more recent ABE variants is not well explored.

1. NG-Cas9 is another SpCas9 variant, which has higher fidelity and flexible PAM. Whether the present variant (V82S/Q154R) could be functional with NG-Cas9?

2. Is there any advantage for the albino/nacre *mitfa*^{W2/W2} zebrafish in vivo screening? Although it is reported previously, we still recommend which A base(s) have been edited and its editing efficiency. The results in Fig 1C should be confirmed with NGS.

3. Based on our experience, ABE8e-SpRY is a variant possessing super-high ABE activity. The authors showed that ABE-Umax-SpRY can target sites such as *rpI9*-NTC that could not be previously edited by ABE8e-SpRY. If it could be applied in human (mice) cells or even plants, additional researchers could be benefited. We highly suggested to extend its application, otherwise, the topic in this manuscript may not be fit. Also, why it is more activate in zebrafish, any clue for its performance?

4. As to Fig 2, because 5'-NGG-3' may not be presented in the target sequence, also, the fidelity of canonical SpCas9 is much lower than NG-Cas9 or even SpRY. A parallel comparison of canonical SpCas9 (WT SpCas9), NG-Cas9 or even SpRY should be performed. Also, the off-target effect (DNA and RNA level) has not been investigated in the whole study.

5. ABE-Umax-rest 2 sacrifices major editing activity, which should be pointed out.

6. We don't know why there is no ABE8e-SpRY control in the Fig 4 and 5. With the present data, we can't be convinced with the advantage of variants identified in this study.

7. We acknowledge that ABE variants were screened with albino/nacre *mitfa*^{W2/W2} zebrafish in vivo, why the authors repeated this study in Fig 6. It is highly recommended with additional gene to confirm the previous findings.

8. We can't get the point from the cited literature, "Several variants of the TadA deaminase domain (e.g. TadA-KR and TadA-9e) have been shown to improve base editing efficiencies in human cells but still suffer from poor performance in vivo". If it is not the case, the author may provide literature to support the scientific question.

Reviewer #2 (Remarks to the Author):

This is an excellent paper that uses the zebrafish model to improve CRISPR-Casp9 adenine base editors in a living organism. This editing system relies on a Cas9 nickase fused to a deaminase, along with a guide RNA to edit the genome and create precise, single nucleotide variants (SNVs). Studies that seek to create models that mimic a SNVs of interest are limited by the target sequence. This paper represents an immense amount of work testing and designing many CRISPR-Casp9 adenine base editors to create SNVs that work with a variety of target sequences, in an in vivo context. This study generates a toolkit or quiver of CRISPR-Cas9 adenine base editors that can be tailored for use based on the target sequence of interest. First the authors screen through the best ABE-variants in vivo in zebrafish and identify one variant, ABE-Umax as the top variant. They then design and select additional variants to increase base editing possibilities. This includes characterization of ABE-variants that work with and without a canonical PAM sequence. Also, ABE-variants that are effective at different locations relative to the protospacer. Lastly ABE-variants that have narrow editing windows to ensure precision in editing. Lastly, this study shows proof of principle base editing of SNVs predicted to be associated with human disease.

The work outlined in this paper will greatly advance research in many in vivo models that seek to use base editors to create new genetic models. In addition, this knowledge may be useful more broadly in gene therapy where base editing is needed to correct deleterious mutations.

The experiments in this manuscript represent an immense amount of in vivo work to systematically define efficient CRISPR-Casp9 base editors that work on distinct target sequences. Overall, the manuscript is extremely well written but could use clarification in several instances. These instances outlined below.

There are a few items that could be clarified or corrected:

Introduction

Include more background on how Cas9 nickases and the deaminases work to repair a DNA template. Summarize relevant ABE/ABE8 with regards to the TadA domain. Also include information about PAM sequences and where deaminases generally edit best. Add info about nucleotide sites and the protospacer. Some of this information is in the results, but add a summary of this information would be helpful in the introduction.

Perhaps additional information about SpRY Cas9 versus more traditional Cas9s

Results

It would be helpful to have more transparency and clarity on how the loci and guides tested were chosen. It seems like the authors are using a subset of loci/guides from previous studies which is fine. But whether this is the case is not clear. Also, if a subset of the previously used loci/guides were used, it is important to know how the subset is chosen-to prevent bias in analyses.

Page 4 The ABE8e variants of the ABE8e variant is confusing.

Also please make it clear that the 10 variants are all based on ABE8e

How did you pick your 10 TadA ABE8e variants? Provide reference (s)

Page 5 Be clearer on the 9 reported loci, with a reference. For these 9 loci, were new guides designed? Or are the guides the same as those already tested in another study? If the same or different, how was the guides chosen?

Page 6 Were the guides and 16 target sites tested the same as those used in the original ABE7.10 study? If the same, why these 16 (of the 28)? If new guides and sites, why?

Page 6 How did you choose your sites to compare ABE-Ultramax and ABE-Umax? ABE-Umax and ABE8e?

Additional comments for results

Page 8 state more clearly how genotyping of the resultant F1 embryos reflected on ABE-Umax-nanos's ability to install base edits? What did you see in the ABEUmax-nanos1 injected F0 vs ABE-Umax injected F0?

Page 11 "were developed through structure guided engineering to have a narrower editing window of 1–2 nucleotides."

Where is this window located? Need some context here.

Page 11 "Finally, we introduced the same mutations into our ABE-Umax-ex1 and ABE-Umaxex2

variants that exhibited more PAM-proximal editing windows.”

-does this refer to the N108Q and N108Q/L145T mutations? If so perhaps these variants should have new names. Especially in Figure 3f.

Page 12/13

Perhaps state that myo7aa and ush1c mutant lateral line hair cells have defects in cationic dye uptake (such as Yo-Pro).

Also, it is misleading to indicate that the startle defects in myo7aa and ush1c mutants are due to dysfunctional lateral line hair cells. The startle defects are primarily due to dysfunctional inner ear hair cells in zebrafish.

Figures, legends and Tables

Figure 1a. More clarity on your diagram. Where is Cas9? Where is the deaminase? Also, the location of ABE8e is misleading on the figure. Is it the blue piece attached to Cas9? Are the ABE's feeding into the systems replacing the blue deaminase? Or do they represent the whole complex? Perhaps change the schematic and add additional information to the legend.

Figure 1d,f; Figure 2a,b State what the A's represents in the figure legend.

Figure 2c Do the heatmaps represent the percentage of successful A to G editing product? This is not 100% clear in the legend, Y-axis or text.

Figure 2c/Supplemental Table 1. The text states that 8 targets from the 26 sites (Figure 2c) were tested for germline transmission. Figure 2c doesn't include adam22, 2 of the sites in Supplementary Table 1.

Figure 2h be clear whether the middle panel a F0 fish injected with ABE-Umax or ABEUmax-nanos1?

Figure 3b it would be nice to have a comparison, even in a supplement to compare these new variants to ABE-UMax.

For Supplemental Tables 4 please add a column to state what Figure and panel (eg Figure 1a,c) the guide and primer pair were used.

Minor

Page 7 extra space 80 %.

Reviewer #3 (Remarks to the Author):

Precise base editing is particularly valuable for disease modeling, allowing patient related single nucleotide variants to be introduced into animals. Previous work (Cornean et al (2022)) demonstrated that the current gold standard adenine base editor Abe8e works well in zebrafish, with efficiencies often over 90%. However, in other cases less robust efficiency and limited targeting opportunities may limit the usefulness of base editing. The authors tested combinations of TadA adenine deaminase domain mutations and a PAM-less nCas9 variant, to develop a new adenine base editor with 68% increased efficiency compared to Abe8e. In fact, the authors demonstrated that base editing is sufficiently productive for use in G0 zebrafish screens, similar to the way that Cas9 is currently applied in 'crispr' screens. They also constructed variants that can target adenines outside the canonical editing window that comprises positions 4-8 in the protospacer, which provides valuable targeting flexibility, especially in order to edit specific adenines while leaving others in the targeting sequence intact. The utility of these variants was demonstrated by the ability to rationally select base editors to selectively edit specific adenines, modifying a range of sites, including splice sites and predicted pathogenic sites from human hearing disorders, leading to defects in hair cell function. Base editing efficiency is rigorously quantified throughout the manuscript. The set of base editors reported here will find wide utility in allowing researchers to address a large range of biological questions, and is suitable for publication.

A slight remaining concern is the frequency of (1) off-target effects (ie, at sequence related genomic loci) and (2) transcriptome-wide A-to-I substitutions. These were not addressed in this paper. Abe8e already has fairly high off-target rates and transcriptome-wide A-to-I edits and it is possible that these may occur at an even higher rate in the new variants, particularly in ABE-Umax-ex1 and -ex2 where the Cas9 domain is modified. For at least a few targets, the authors should select several predicted off-target sites and check whether off-target editing occurs, or even just analyze their existing NGS data. This obviously most important for G0 screens where additional mutations can't be removed through crossing. Second, the authors should analyze the transcriptome-wide rate of A to I conversions in RNAseq data.

Other than that I have no reservations – my comments below are mainly intended to help the authors to clarify several aspects of their methods and figures by providing more detailed information.

1. The notion that the base editor has a specific range is nicely presented and quantified in 2c. It would help the non-specialist reader to briefly describe the editing window of Abe8e before presenting results, and point out that this can present a technical obstacles to its use (this appears on p9, I think it should be earlier in the manuscript).

2. Why is data for specific adenosines reported or excluded? For example, in 1d, for rpl17-NTG the targeting sequence is GGCTAACCAAGTACCTGAAGG. There is quantification for A5 and A6. What about the other adenosines, for example A8, which is within the editing range? Same question pertains to other targets throughout the manuscript. What is the criterion for reporting or not reporting editing for various in-range targets?

3. In 1d and 1f, the organization of the columns along the x-axis seems arbitrary. For 1d, it would seem most natural to cluster targets associated with each gene together (for example, which helps to show that rpl9 is generally a tough target). For 1f, ordering could be based on PAM sequence. There should be some rationale for the order in these graphs.

4. Supp Fig 1. Please define XTEN and bpNLS. Similarly Fig 3a, define ART-linker.

5. 1d (and many other places). It would be useful to briefly mention in the main text how editing efficiency was calculated, including the sequencing method. I inferred use of Sanger sequencing with analysis using EditR. Also, Supp Fig 3 confirmed that this approaches gives similar results to NGS - the authors could consider moving Supp Fig 3 to earlier in the manuscript so reader is assured of the rigor of their quantification method from the outset.

6. Methods: please define "MS modifications"

7. For germline transmission rates, what exactly is measured in "germline editing efficiency" ? I think it is the percent of founders, any of whose progeny show transmission. If so, this metric depends on how many embryos were screened for each founder. For example for the 100% target itsn1-T3, perhaps more embryos were screened than for other founders. What is the average proportion of editing embryos for each positive founder?

8. p8 tubb4b, please mention briefly what disease this is a model for.

9. Figure 3b is a nice summary, but the underlying efficiency measures at the 7 sites should also be presented in supplementary data.

10. The color scale in 3d makes it hard to compare efficiencies - the scale in 2c is more suitable.

11. The innovation of shifting the editing range from 4-8 to 3-6 or 12-15 is an important contribution, but given very short shrift. The results are only schematically presented in figure 3f and shown without quantification in Supp Fig 6. These data should be fully presented in a similar form to Figure 3b.

12. p14. The base edits in cdh23 and tyr both lead to cryptic splice changes. This is certainly a possible (even likely) outcome, but equally, intron retention or exon skipping could have occurred. So the statement "We confirmed that ABE-Umax toolkit treatment resulted in the expected pre-mRNA splicing defects" could be expressed slightly differently, and the authors might explicitly mention that both changes lead to cryptic splicing. The schematics of the changes in 5g and 5h are not very clear. There are supposed to be 15 and 13 bp deletions respectively, but only 2 and 5 nucleotides are shown in the boxed area.

13. Figure 6b/c are inelegant and take some time to interpret (for example, it adds confusing clutter to add the precise sequence of both DNA strands). This detailed information should be in the supplement and the replaced in the main text by a schematized simple figure showing where the guides align on the exon with respect to the targeted base.

14. The experiment in Figure 6 is very helpful in that it demonstrates how an ABE-umax variant can be rationally selected. I suggest that the authors augment this by providing a summary figure, showing how the various new editors could be used for a hypothetical target along with the mean editing efficiency of each.

We express our gratitude to the editor for the opportunity to revise the manuscript and the reviewers for their insightful feedback. We revised the manuscript accordingly, and added one new main figure, modified five figures, and added six new supplementary figures. The following specific answers to each question addresses concerns raised by the reviewers. We believe these improvements significantly strengthen the manuscript and make it suitable for publication in Nature Communications. All changes are highlighted in the revised manuscript.

Reviewer 1	1
Reviewer2	4
Reviewer 3	12

Reviewer 1

The authors utilized albino/nacre mitfaW2/W2 zebrafish to optimize ABE. They showed that, within the ABE-Umax platform, a shifted, narrowed, or broadened editing windows, reduced bystander mutation frequency, as well as highly flexible PAM sequence requirements could be achieved. The study is well designed, and the new base editing tool may be of broad use for zebrafish genetic engineering. While its advantage over more recent ABE variants is not well explored.

Response: We'd like to express our sincere thanks to the reviewer for their encouraging feedback. Recognizing the importance of our work truly motivates us.

1. NG-Cas9 is another SpCas9 variant, which has higher fidelity and flexible PAM.

Response: Thank you for your suggestion. NG-Cas9 is indeed a SpCas9 variant with higher fidelity and a flexible PAM. However, the SpRY variant used in this study has even more flexible PAMs (PAM=NNN) compared to the NG-Cas9 variant (PAM=NG). Regarding fidelity, both our previous study (Fig. 4e, PMID: 35701400) and this study (Supplementary Figure 12) confirmed that the SpRY variant does not exhibit apparent off-target effects at the DNA level in zebrafish. Therefore, we do not have evidence that NG-Cas9 offers a significant advantage in this manuscript. Since all the experiments in the current manuscript were performed with the SpRY variant, we only present data related it. We are studying the similarities and differences in PAM editing efficiency among different Cas9 variants. It does require extensive analysis beyond the scope of this manuscript and will be presented in a future publication.

Whether the present variant (V82S/Q154R) could be functional with NG-Cas9?

Response: We performed additional experiments, and the data suggest that the variant (V82S/Q154R) is functional with NG-Cas9 (**Additional Figure for Reviewer below, see blue bar chart**).

Additional Figure 1 for reviewer. Comparison of ABE-Umax-NG-cas9, ABE-Umax-X-cas9, and ABE-Umax-SpRY Editing Efficiency at NGN PAM Target Sites in Zebrafish

- **(a) Quantification of A-G conversion efficiency:** A-to-G conversion frequencies at each adenine within the 20 bp protospacer were measured using EditR (three independent experiments).
- **(b) Comparative assessment of editing efficiency:** Mean editing efficiencies of ABE-Umax-NG-cas9, ABE-Umax-Xcas9, and ABE-Umax-SpRY were analyzed using a plot constructed from the data in (a). Each data point denotes the average editing activity at a specific site. The graph's central dotted line indicates the mean across all data points. P-values are displayed within the violin plot.

2. Is there any advantage for the albino/nacre *mitfa*^{W2/W2} zebrafish in vivo screening? Although it is reported previously, we still recommend which A base(s) have been edited and its editing efficiency. The results in Fig 1C should be confirmed with NGS.

Response: Thank you for your recommendation. *mitfa*^{W2/W2} zebrafish are indeed a very well-established model for visual genetic screening and imaging as this strain offers a quick, visible readout, allowing us to easily and rapidly assess the editing activity of different variants based on the degree of rescued phenotype (pigment recovery). Even a single black melanin pigment point is discernible against an albino background.

To address concerns about NGS validation, we randomly selected 90 embryos (30 per group) for pooled sequencing. These results confirm that the rescued phenotypes correlate with T to C (A to G on the reversed strand) base conversions, and the efficiency of ABE-Umax was three times that of ABE8e. Overall, base editor activity aligns consistently with our sequencing data. **(Please see Supplementary Figure 2).**

3. Based on our experience, ABE8e-SpRY is a variant possessing super-high ABE activity. The authors showed that ABE-Umax-SpRY can target sites such as rpl9- NTC that could not be previously edited by ABE8e-SpRY. If it could be applied in human(mice) cells or even plants, additional researchers could be benefited. We high suggested to extend its application, otherwise, the topic in this munuscript may not be fit. Also, why it is more activate in zebrafish, any clue for its performance?

Response: Thank you for recognizing the broader potential of ABE-Umax-SpRY. As rightfully illustrated on Ensembl and UCSC genome browsers, the popular zebrafish genome and broad scientific community are extremely well established and require tools of their own for vertebrate biology and translational studies. While ABE8e exhibits high activity in *in vitro* systems, we sought variants with improved performance in zebrafish for human disease modeling. Since ABE8e demonstrates limited activity in zebrafish, our goal was to identify a variant offering a significant advantage in generating biallelic variants in zebrafish. Our screen identified Umax

variants as the most efficient in zebrafish. We hypothesize they will also be active in other fish species and potentially other species. However, due to the study's focus on developing a highly efficient zebrafish variant, testing across additional species falls outside the scope of this manuscript. We are, however, open to collaborating with other labs to explore Umax variant activity in other models following manuscript publication.

4. As to Fig 2, because 5'-NGG-3' may not be presented in the target sequence, also, the fidelity of canonical SpCas9 is much lower than NG-Cas9 or even SpRY. A parallel comparison of canonical SpCas9(WT SpCas9), NG-Cas9 or even SpRY should be performed. Also, the off-target effect (DNA and RNA level) has not been investigated in the whole study.

Response: Thank you for your suggestion. We have addressed this concern by comprehensively analyzing off-target effects of ABE tools in zebrafish at both the DNA and RNA levels. New data is presented in new Figure 4, new supplementary figures 9-12, and additional details have been incorporated into the manuscript (pages 13-14). Our data show that the off-target effects of the new variants are comparable to those of ABE8e variants. Regarding the NG-Cas9 variant question, please refer to our response to the first question.

5.ABE-Umax-rest 2 sacrifices major editing activity, which should be pointed out.

Response: Yes, we agree with this reviewer, and we have now added this sentence "As ABE-Umax-rest2 exhibited an apparent reduction in on-target activity, we recommend ABE-Umax-rest1 for precise editing." in the revised manuscript (page12).

6.We don't know why there is no ABE8e-SpRY control in the Fig 4 and 5. With the present data, we can't be convinced with the advantage of variants identified in this study.

Response: A previous paper by the co-first author Dr. Liang demonstrated the activity of ABE 8e-SpRY (Liang et al Nature Communications PMID: 35701400). This novel work prioritizes data-driven evidence to showcase ABE-Umax's superiority over ABE8e. Figures 1d and 2a provide extensive data (sites) supporting this claim. Figures 5 and 6 (previously Figures 4 and 5) demonstrate the practical applications of ABE-Umax and ABE-Umax-2, and both tools and all targeted sites utilize the canonical "NGG" PAM sequence. Therefore, comparing them with ABE8e-SpRY (specific for "NNN" PAM) would not be a correct comparison. Additionally, figure 2a demonstrates that ABE-Umax-SpRY exhibits lower activity compared to ABE-Umax at "NGG" PAM sites.

7. We acknowledge that ABE variants were screened with albino/nacre *mitfa*^{W2/W2} zebrafish in vivo, why the authors repeated this study in Fig 6. It is highly recommended with additional gene to confirm the previous findings.

Response: Thank you for your suggestion. We understand the reviewer's concern regarding the use of the same gene (*mitfa*) in Figures 1 and 6 (now 7). However, the purposes of these experiments are distinct.

- In Figure 1, we employed the existing *mitfa*^{W2/W2} mutant zebrafish line to allow for rapid and visual assessment of editing efficiency by observing black melanin pigment rescue.
- In Figure 6 (now 7), it is the opposite. Our objective was to introduce in a wildtype pigmented background, a disease-relevant mutation (S357P) into the zebrafish MITF gene to mimic the Waardenburg syndrome, a genetic disorder causing hearing loss and

pigmentation changes. This mutation is novel in animal models, and our study represents the first *in vivo* validation.

The key message remains the importance of selecting the most appropriate editing tool based on the specific target site, rather than solely focusing on the gene itself.

8. We can't get the point from the cited literature, "Several variants of the TadA deaminase domain (e.g. TadA-KR and TadA-9e) have been shown to improve base editing efficiencies in human cells but still suffer from poor performance in vivo". If it is not the case, the author may provide literature to support the scientific question.

Response: We apologize for any confusion here. We meant that there have been new variants of TadA, and these variants' efficiency still limited or has not been tested in vivo. We used these variants to synergistically enhance ABE efficiency *in vivo* and text has been changed as follows: "Recent studies have shown improved editing efficiencies in human cells using variants of the TadA deaminase domain (e.g., TadA-KR and TadA-9e), however, their in vivo efficiencies remain limited or has not been tested". We have moved this sentence to the introduction part.

Reviewer2

This is an excellent paper that uses the zebrafish model to improve CRISPR-Casp9 adenine base editors in a living organism. This editing system relies on a Cas9 nickase fused to a deaminase, along with a guide RNA to edit the genome and create precise, single nucleotide variants (SNVs). Studies that seek to create models that mimic a SNVs of interest are limited by the target sequence. This paper represents an immense amount of work testing and designing many CRISPR-Casp9 adenine base editors to create SNVs that work with a variety of target sequences, in an *in vivo* context. This study generates a toolkit or quiver of CRISPR-Cas9 adenine base editors that can be tailored for use based on the target sequence of interest. First the authors screen through the best ABE-variants in vivo in zebrafish and identify one variant, ABE-Umax as the top variant. They then design and select additional variants to increase base editing possibilities. This includes characterization of ABE-variants that work with and without a canonical PAM sequence. Also, ABE-variants that are effective at different locations relative to the protospacer. Lastly ABE-variants that have narrow editing windows to ensure precision in editing. Lastly, this study shows proof of principle base editing of SNVs predicted to be associated with human disease.

The work outlined in this paper will greatly advance research in many *in vivo* models that seek to use base editors to create new genetic models. In addition, this knowledge may be useful more broadly in gene therapy where base editing is needed to correct deleterious mutations. The experiments in this manuscript represent an immense amount of *in vivo* work to systematically define efficient CRISPR-Casp9 base editors that work on distinct target sequences. Overall, the manuscript is extremely well written but could use clarification in several instances. These instances outlined below.

Response: We extend our heartfelt appreciation to your constructive feedback, which greatly motivates us to further advance our research.

1. Include more background on how Cas9 nickases and the deaminases work to repair a DNA template. Summarize relevant ABE/ABE8 with regards to the TadA domain. Also include information about PAM sequences and where deaminases generally edit best. Add info about nucleotide sites and the protospacer. Some of this information is in the results but add a summary

of this information would be helpful in the introduction. Perhaps additional information about SpRY Cas9 versus more traditional Cas9s

Response: Thanks for your suggestion. We have added additional information in the introduction of revised manuscript as follows on page 3-4.

“Cas9, complexed with a single guide RNA (sgRNA), initiates the formation of an R-loop structure at the target DNA site. Adenine deaminase fused with Cas9 nickase then converts the exposed adenine to inosine. During DNA replication, inosine is misread as guanosine, resulting in a permanent A-to-G base change. Additionally, Cas9 nickase (nCas9) introduces a nick in the complementary strand, prompting the DNA repair machinery to preferentially use the edited strand as a template, further boosting editing efficiency. ABE8e, the most widely used ABE variant due to its efficiency and compatibility with a wide range of model systems, typically exhibits an editing window spanning positions 4-8 relative to the protospacer adjacent motif (PAM) sequence on the target DNA strand. However, its targeting efficiency remains low at some specific sites, and its constrained editing window and preference for NGG PAM severely limit the sequence space that can be targeted by existing base editors. Recent studies have shown improved editing efficiencies in human cells using variants of the TadA deaminase domain (e.g., TadA-KR and TadA-9e). Additionally, novel ABE tools with activity in shifted targeting windows have been developed. Furthermore, researchers have engineered Cas9 variants with superior efficiency, fidelity, and diverse PAM preferences. Notably, the SpRY variant recognizes a broader "NNN" PAM and demonstrates efficient, near PAM-independent genome editing in zebrafish”.

2. It would be helpful to have more transparency and clarity on how the loci and guides tested were chosen. It seems like the authors are using a subset of loci/guides from previous studies which is fine. But whether this is the case is not clear. Also, if a subset of the previously used loci/guides were used, it is important to know how the subset is chosen-to prevent bias in analyses.

Response:

While we randomly selected some target sites, we prioritized genes relevant to our ongoing functional work due to the high cost of synthetic gRNAs. We also utilized previously published targets. The focus of our lab on hearing loss and neurodevelopmental genes naturally led us to select loci that we are interested in functionally characterizing. Throughout the study, we systematically evaluated over 70 sites (largest number tested in zebrafish thus far), all demonstrating robust activity regardless of selection method. This highlights a key advantage of our tool. Additionally, selected targets are from the genes covering 19/25 chromosomes **(Additional Figure for Reviewer 2, see below)**

For transparency, we have added specific examples within the text to illustrate our target selection process:

1. Page 6: Comparison of targeting efficiencies between ABE-Umax-SpRY and ABE8e-SPRY at 9 previously reported loci (reference 19).
2. Page 7: Characterization of ABE-Umax-SPRY editing profiles across 16 endogenous zebrafish target sites spanning 10 genes of interest, including sites with diverse PAM sequences (NRN and NYN).
3. Page 7: Creation of ABE-Ultramax (ABE-Umax) and comparison of its editing efficiency at 8 reported genomic sites with "NGG" PAM sequences (references 20, 26).

4. Page 8: Analysis of an additional 28 endogenous zebrafish target sites with NGG PAM sequences across 17 genes associated with neurodevelopmental disorders, and hearing loss genes.

Additional Figure 2 for Reviewer: Distribution of genes studied in this manuscript in the zebrafish genomes.

3. Page 4 The ABE8e variants of the ABE8e variant is confusing. Also please make it clear that the 10 variants are all based on ABE8e How did you pick your 10 TadA ABE8e variants? Provide reference (s).

Response: We appreciate your suggestion for a better description of the 10 TadA variants. We have revised **Supplementary Figure 1** to now showcase the genotypes of all reported TadA variants, including tested in this manuscript. Additionally, we have added a new description and references for this section (references 11, 12, 22, 23). Since existing TadA variants were developed by various groups using different strategies (references 11, 12, 22, 23), we hypothesized that combining specific mutations from individual TadA and Cas9 variants could synergistically improve base editing efficiency and specificity. To test this, we designed ten new TadA variants by incorporating different permutations of previously validated mutations (see New Supplementary Figure 1 below).

a

b

Position	23	27	36	47	48	51	76	82	84	106	108	109	111	119	122	123	127	146	147	149	152	154	155	156	157	166	167
WT TadA	W	E	H	R	P	R	I	V	L	A	D	A	T	D	H	H	N	S	D	F	R	Q	E	I	K	T	D
TadA7.10	R		L		A	L			F	V	N						Y		C	Y		P		V	F	N	
TadA8.17	R		L		A	L		S	F	V	N						Y		C	Y		P	R	V	F	N	
TadA8.20	R		L		A	L	Y	S	F	V	N								C	R		P	R	V	F	N	
TadA8e	R		L		A	L			F	V	N	S	R	N	N	Y		C		Y	P		V	F	N	I	N
TadA-8eKR	R		L		A	L			F	V	N	S	R	N	N	Y	K	C		Y	P	R	V	F	N	I	N
TadA-9e	R		L		A	L			F	V	N	S		N	N	Y	K	C		Y	P	R	V	F	N	I	N
variant1	R		L		A	L		S	F	V	N	S	R	N	N	Y		C		Y	P		V	F	N	I	N
variant2	R		L		A	L			F	V	N	S	R	N	N	Y		C		Y	P	R	V	F	N	I	N
variant3	R		L		A	L		S	F	V	N	S	R	N	N	Y		C		Y	P	R	V	F	N	I	N
variant4	R		L		A	L		S	F	V	N	S	R	N	N	Y		C	R	Y	P	R	V	F	N	I	N
variant5	R		L		A	L		S	F	V	N	S	R	N	N	Y		C		Y	P	R	V	F	N	I	N
variant6	R		L		A	L		S	F	V	N	S	R	N	N	Y	K	C		Y	P	R	V	F	N	I	N
variant7	R		L		A	L			F	V	N	S		N	N	Y	K	C		Y	P	R	V	F	N	I	N
variant8	R		L		A	L	Y	S	F	V	N	S	R	N	N	Y		C		Y	P	R	V	F	N	I	N
variant9	R		L		A	L		S	F	V	N	S		N	N	Y	K	C		Y	P	R	V	F	N	I	N
variant10	R		L		A	L	Y	S	F	V	N	S	R	N	N			C	R	Y	P	R	V	F	N	I	N

New Supplementary Figure 1: Genotypes of Reported and New 10 TadA Variants

(a): Schematic diagrams of the adenine base editors used in this screen **XTEN**: Peptide linker sequence (SGSETPGTSESATPES) connecting the nucleoside deaminase domain and nCas9 domain. **bpNLS**: Bipartite nuclear localization signals.

(b): Mutations in TadA variants. **Gray**: Mutations present in TadA7.10. **Orange, Green, Cyan, Pink, Purple**: Additional mutations reported in TadA8.17, TadA8.20, TadA8e, TadA8eKR, and TadA9e, respectively. **Red**: Mutations introduced in our 10 TadA variants (based on TadA8e).

4. Page 5 Be clearer on the 9 reported loci, with a reference. For these 9 loci, were new guides designed? Or are the guides the same as those already tested in another study? If the same or different, how was the guides chosen?

Response: Thank you for your suggestion. We have added the reference (Liang et al Nature Communications PMID: 35701400) to clarify that the guide RNAs (gRNAs) used are not newly designed. They are the same gRNAs employed in our previous study.

Our target site selection process was guided by two key factors:

- **Cost-Effectiveness:** Synthesizing new gRNAs can be expensive. Therefore, we prioritized existing, well-tested gRNAs previously published in the literature for our experiments.
- **Research Focus:** Our lab focuses on understanding the function of hearing and neurodevelopmental genes. Consequently, many target loci were chosen because they align with our ongoing research interests in these areas.

While we primarily utilized existing gRNAs, it is important to note that we also systematically evaluated over 70 additional target sites throughout the study. These additional sites, regardless of selection method, displayed robust activity with our tool, highlighting its broad applicability.

5. Page 6 Were the guides and 16 target sites tested the same as those used in the original ABE7.10 study? If the same, why these 16 (of the 28)? If new guides and sites, why?

Response: The 16 target sites tested in this study differ from those used in the original ABE7.10 study. In ABE7.10, we employed guide RNAs (gRNAs) with a specific "NGG" PAM sequence. In contrast, all 16 target sites in this study possess non-canonical PAM sequences (sequences other than "NGG"). This selection of gRNAs aligns with the criteria mentioned previously in responses to questions 2 and 4.

6. Page 6 How did you choose your sites to compare ABE-Ultramax and ABE-Umax? ABE-Umax and ABE8e?

Response: We apologize for any confusion caused by the initial description of our target sites. All eight sites were chosen based on previous research (PMID: 30458760; PMID: 35701478). The editing activity of these sites with existing tools was variable:

- Two sites showed reported A-to-G editing with ABE7.10 in zebrafish.
- Four sites displayed no A-to-G editing using ABE7.10.
- Two sites demonstrated C-to-T editing with CBE-SpRY, but their A-to-G editing potential remained untested.

These references have been added to the manuscript for clarity (PMID: 30458760; PMID: 35701478).

Furthermore, to enhance the comparison of different editing tools at the same locus, we have merged Figures 2a and 2b. This new Figure 2 provides a unified comparison of editing efficiency across various tools.

7. Additional comments for results.

Page 8 state more clearly how genotyping of the resultant F1 embryos reflected on ABE-Umax-nanos's ability to install base edits? What did you see in the ABEUmax-nanos1 injected F0 vs ABE-Umax injected F0?

Response: We appreciate your suggestion. We have incorporated it by revising the sentence to: "Genotyping of outcrossed F1 embryos from breeding of the ABE-Umax-nanos1 F0 founder with AB wild type fish demonstrated the tool's ability to introduce transmissible base edits in essential genes". Additionally, we have included a dedicated "zebrafish screening" section in the methods section (pages 24-25) for clarity. We have presented the phenotypic data in Figure 2f and 2g. The text describing them has been revised as follows:

"While developing a patient-relevant disease model using ABE-Umax, we discovered that the Q11R mutation in the *tubb4b* gene resulted in body curvature, hydrocephalus, and heart edema in all injected embryos (Figure 2d, e). This limited their survival to 7 days post-fertilization (dpf). Conversely, embryos injected with ABE-Umax-nanos1 targeting the *tubb4b* site exhibited normal development and could be raised to adulthood for breeding (Figure 2f)."

8. Page 11 "were developed through structure guided engineering to have a narrower editing window of 1–2 nucleotides." Where is this window located? Need some context here.

Response: We have modified the sentence as follows: “Recently, two ABE variants, ABE8e-N108Q and ABE9 (N108Q+L145T), were developed through structure-guided engineering to have a narrower editing window of 1–2 nucleotides at protospacer positions 5 or 6 (counting the PAM as positions 21–23).”

9. Page 11 “Finally, we introduced the same mutations into our ABE-Umax-ex1 and ABE-Umaxex2 variants that exhibited more PAM-proximal editing windows.”

-does this refer to the N108Q and N108Q/L145T mutations? If so perhaps these variants should have new names. Especially in Figure 3f.

Response: Thank you for pointing this inaccuracy. We have modified the sentence as follows: “As ABE-Umax-rest2 exhibited a significant reduction in on-target activity, we recommend ABE-Umax-rest1 for precise editing. To achieve narrower editing windows, we further introduced the N108Q mutation into ABE-Umax-ex1 and ABE-Umax-ex2 variants, which initially displayed broader PAM-proximal editing. These modified variants, named ABE-Umax-ex1-rest1 and ABE-Umax-ex2-rest1, demonstrated a remarkable ability to refine the editing window to target positions 4-6 and 12-15, respectively (Figure 3e).” for figure 3f, we have revised as the follows:

Figure 3(f). A schematic diagram illustrates the editing window for ABE-Umax tools. Editing preferences are indicated by a red shading gradient, with the darkest red marking the most preferred position and lighter shades representing less preferred edits. A blue line highlights the precise editing position. The red triangle denotes the theoretical SpCas9 cutting site, while the PAM sequence and its complement are shown in green.

10. Page 12/13 Perhaps state that myo7aa and ush1c mutant lateral line hair cells have defects in cationic dye uptake (such as Yo-Pro).

Response: Per your suggestion, we have modified the sentence as follows: “While control animals exhibited normal Yo-Pro-1 uptake by lateral hair cells, both base-edited variants failed to uptake the dye. This indicates a disruption of lateral hair cell function, which we attribute to the loss of function in either *myo7aa* or *ushc1c* genes.”

11. Also, it is misleading to indicate that the startle defects in myo7aa and ush1c mutants are due to dysfunctional lateral line hair cells. The startle defects are primarily due to dysfunctional inner ear hair cells in zebrafish.

Response: Thank you for pointing out this inaccuracy. We have modified the sentence as follows: “Previous studies have shown that zebrafish with loss-of-function in *myo7aa* or *ush1c* have dysfunctional hair cells composing their inner ear hair cell and lateral line hair cells, resulting in a measurable loss of the typical startle response triggered by sharp vibrations in the water environment.”

12. Figure 1a. More clarity on your diagram. Where is Cas9? Where is the deaminase? Also, the location of ABE8e is misleading on the figure. Is it the blue piece attached to Cas9? Are the ABE's feeding into the systems replacing the blue deaminase? Or do they represent the whole complex? Perhaps change the schematic and add additional information to the legend.

Response: Thank you for your suggestion. We have updated figure 1a.

Figure 1

(a) Schematic diagram of the screening process for efficient adenine base editing tools. Abbreviations: Cas9n (Cas nickase), TadA (adenine deaminase), PAM (protospacer adjacent motif), sgRNA (single guide RNA). Different colors (blue, yellow, purple, pink, and green) represent various TadA variants used in the deaminases.

13. Figure 1d,f; Figure 2a,b State what the A's represents in the figure legend.

Response: Thank you for your suggestion, it has been corrected. All related figure legend contained this sentence: "The position of the editing base in the gRNA was labelled with numbers".

14. Figure 2c Do the heatmaps represent the percentage of successful A to G editing product? This is not 100% clear in the legend, Y-axis or text.

Response: Thank you for your suggestion. We have updated the figure and legend for the heat map that displays the average A to G editing efficiency of ABE-Umax across 28 targets at different position sites.

Figure 2 (c). Heatmap depicting the average A-to-G editing efficiency of ABE-Umax across 28 target sites. Editing efficiency is represented by a color gradient, with red indicating 100% efficiency and blue indicating 0% efficiency. The black dotted line highlights the preferred editable range of ABE-Umax, between positions 4 and 8.

15. Figure 2c/Supplemental Table 1. The text states that 8 targets from the 28 sites (Figure 2c) were tested for germline transmission. Figure 2c doesn't include adam22, 2 of the sites in Supplementary Table 1.

Response: Thank you for pointing this mistake. We have added these two sites into figure 2c.

16. Figure 2h be clear whether the middle panel a F0 fish injected with ABE-Umax or ABEUmax-nanos1?

Response: Thank you for pointing this inaccuracy. We have updated the middle panel in figure2h.

17. Figure 3b it would be nice to have a comparison, even in a supplement to compare these new variants to ABE-Umax.

Response: Thank you for your suggestion. We have added Supplementary Figure 6 to show this data.

Supplementary Figure 6. Assessment of the editing efficiency and targeting window of ABE-Umax-ex1 and ABE-Umax-ex2 using EditR to quantify A-to-G conversion frequencies at each adenine nucleotide within a 20-bp protospacer sequence. Three independent experiments were conducted and editing frequencies > 0.20 are highlighted in the heatmap. The color scale ranges from blue (100% efficiency) to white (0% efficiency).

18. For Supplemental Tables 4 please add a column to state what Figure and panel (eg Figure 1a,c) the guide and primer pair were used.

Response: Thank you for your suggestion, this part has been added.

Page 7 extra space 80 %.

Response: This has been corrected. Thank you!

Reviewer 3

Precise base editing is particularly valuable for disease modeling, allowing patient related single nucleotide variants to be introduced into animals. Previous work (Cornean et al (2022)) demonstrated that the current gold standard adenine base editor Abe8e works well in zebrafish, with efficiencies often over 90%. However, in other cases less robust efficiency and limited targeting opportunities may limit the usefulness of base editing. The authors tested combinations of TadA adenine deaminase domain mutations and a PAM-less nCas9 variant, to develop a new adenine base editor with 68% increased efficiency compared to Abe8e. In fact, the authors demonstrated that base editing is sufficiently productive for use in G0 zebrafish screens, similar to the way that Cas9 is currently applied in 'crispant' screens. They also constructed variants that can target adenines outside the canonical editing window that comprises positions 4-8 in the protospacer, which provides valuable targeting flexibility, especially in order to edit specific adenines while leaving others in the targeting sequence intact. The utility of these variants was demonstrated by the ability to rationally select base editors to selectively edit specific adenines, modifying a range of sites, including splice sites and predicted pathogenic sites from human hearing disorders, leading to defects in hair cell function. Base editing efficiency is rigorously quantified throughout the manuscript. The set of base editors reported here will find wide utility

in allowing researchers to address a large range of biological questions, and is suitable for publication.

Response: We sincerely appreciate your supportive feedback.

1. A slight remaining concern is the frequency of (1) off-target effects (ie, at sequence related genomic loci) and (2) transcriptome-wide A-to-I substitutions. These were not addressed in this paper. Abe8e already has fairly high off-target rates and transcriptome-wide A-to-I edits and it is possible that these may occur at an even higher rate in the new variants, particularly in ABE-Umax-ex1 and -ex2 where the Cas9 domain is modified. For at least a few targets, the authors should select several predicted off-target sites and check whether off-target editing occurs, or even just analyze their existing NGS data. This obviously most important for G0 screens where additional mutations can't be removed through crossing. Second, the authors should analyze the transcriptome-wide rate of A to I conversions in RNAseq data. Other than that I have no reservations – my comments below are mainly intended to help the authors to clarify several aspects of their methods and figures by providing more detailed information.

Response: Thank you for your suggestion. In the revised manuscript, we have thoroughly analyzed the off-target effects of ABE tools in zebrafish, considering both DNA and RNA levels. For detailed information, please refer to Figure 4, Supplementary Figures 10-13, and the additional text on pages 13 and 14.

Figure 4. Off-target analysis of ABE-Umax-related tools in zebrafish

a) Potential off-target sites at the tyr-g4 locus. The top three high-scoring off-target sites are shown. Mismatched bases are indicated in lowercase, and the potentially editable A is highlighted in red.

b) DNA off-target comparison of ABE78e and ABE-Umax at the tyr-g4 locus. Editing efficiencies are displayed with error bars indicating mean \pm s.d. ($n = 3$ biological replicates).

c) Transcriptome analysis of edited adenine nucleotides in zebrafish embryos. Embryos were injected with ABE8e+tyr-g4, ABE-Umax-ex1+tyr-g4, ABE-Umax-ex2+tyr-g4, or their related mRNA only. Data from two independent replicates are shown.

d) RNA A-to-I conversion frequencies in injected zebrafish embryos. Representative jitter plots display frequencies for embryos injected with ABE8e+tyr-g4, ABE-Umax-ex1+tyr-g4, ABE-Umax-ex2+tyr-g4, or their related mRNA only. Data from two independent replicates are shown.

2. The notion that the base editor has a specific range is nicely presented and quantified in 2c. It would help the non-specialist reader to briefly describe the editing window of Abe8e before

presenting results and point out that this can present a technical obstacle to its use (this appears on p9, I think it should be earlier in the manuscript).

Response: Thank you for your suggestion! We have incorporated this information into the introduction of our revised manuscript as follows:

“ABE8e, the most widely used ABE variant due to its efficiency and compatibility with a wide range of model systems, typically exhibits an editing window spanning positions 4-8 relative to the protospacer adjacent motif (PAM) sequence on the target DNA strand. However, its targeting efficiency remains low at some specific sites, and its constrained editing window and preference for NGG PAM severely limit the sequence space that can be targeted by existing base editors.” (on page 4)

3. Why is data for specific adenosines reported or excluded? For example, in 1d, for rpl17-NTG the targeting sequence is GGCTAACAAGTACCTGAAGG. There is quantification for A5 and A6. What about the other adenosines, for example A8, which is within the editing range? Same question pertains to other targets throughout the manuscript. What is the criterion for reporting or not reporting editing for various in-range targets?

Response: Thank you for highlighting the inaccuracy. We have thoroughly reviewed our manuscript and now present the adenine editing efficiency at all loci where the editing frequency exceeds 0.20. Positions without values indicate an editing efficiency of 0.

4. In 1d and 1f, the organization of the columns along the x-axis seems arbitrary. For 1d, it would seem most natural to cluster targets associated with each gene together (for example, which helps to show that rpl9 is generally a tough target). For 1f, ordering could be based on PAM sequence. There should be some rationale for the order in these graphs.

Response: Thank you for your suggestion. We have improved these figures as suggested.

Figure 1(d). Editing efficiency comparison between ABE8e-SpRY and ABE-Umax-SpRY using 9 gRNAs targeting NNN PAMs. The edited base position within each gRNA is indicated by numbers. Data is presented as mean \pm standard deviation (SD) calculated from three biological replicates.

Figure1f (original manuscript), now figure 1(e). A heat map showing the average A-to-G editing efficiency of ABE-Umax-SpRY across 16 targets at different positions. Editing efficiency (A to G) is represented by a color gradient from red (100%) to blue (0%). A-to-G conversion frequencies at each adenine nucleotide within the 20 bp protospacer were quantified using EditR. Three independent experiments were performed and editing frequencies exceeding 0.20 are labeled on the heat map. The black dotted line highlights ABE-Umax-SpRY's preferential editing range, spanning positions 4 through 8.

5. Supp Fig 1. Please define XTEN and bpNLS. Similarly, Fig 3a, define ART-linker.

Response: Thank you for your suggestion. We've incorporated the information into the relevant figure legends. Here's a clarification of the terms used:

- **XTEN:** A peptide linker composed of the amino acid sequence SGSETPGTSESATPES, bridging the nucleoside deaminase and nCas9 domains.
- **bpNLS:** Stands for bipartite nuclear localization signals.
- **ART-linker (and similarly, GSSGSS-linker):** These represent abbreviations for amino acid sequences linking the C-terminus of *tadA* variants and Cas9. We used the same ART-linker sequence as reported in the previous paper (PMID: 33203850).
- For clarity, we replaced 'ART-linker' with 'ART amino acids' in Fig3a.

Figure 3

Figure 3 (a). Schematics showing constructs of ABE-Umax-ex1 and ABE-Umax-ex2 designed to shift the editing window of adenine base editing.

6. 1d (and many other places). It would be useful to briefly mention in the main text how editing efficiency was calculated, including the sequencing method. I inferred use of Sanger sequencing with analysis using EditR. Also, Supp Fig 3 confirmed that this approaches gives similar results

to NGS - the authors could consider moving Supp Fig 3 to earlier in the manuscript so reader is assured of the rigor of their quantification method from the outset.

Response: To improve clarity for readers, we have added a sentence to the first results section: "We employed the established EditR software for quantitative analysis of Sanger sequencing data to streamline and expedite the evaluation of base conversion efficiency. This approach has been previously described (PMID: 31021262)

7. Methods: please define "MS modifications"

Response: Thank you for your suggestion. MS modifications means MS (2'-O-methyl (M), 2'-O-methyl 3'phosphorothioate) modifications. We have added this information in the methods.

8. For germline transmission rates, what exactly is measured in "germline editing efficiency" ? I think it is the percent of founders, any of whose progeny show transmission. If so, this metric depends on how many embryos were screened for each founder. For example for the 100% target itsn1-T3, perhaps more embryos were screened than for other founders. What is the average proportion of editing embryos for each positive founder?

Response: We apologize for not clarifying it in detail. We have addressed this by adding a dedicated "Zebrafish Screening" section in the Methods section (pages 24-25):

“Screening for Germline Transmission Events: To identify embryos with successful germline transmission of the edited locus, we collected two groups (5 embryos per group) of sibling embryos at 2 days post-fertilization (2 dpf) from F1 generation fish. Genomic DNA was extracted from the collected embryos, and the targeted locus was examined for base editing events using PCR and Sanger sequencing. The germline targeting efficiency was calculated as the percentage of positive founders (showing base editing) among all screened founders. For each positive founder fish, we evaluated the transmission rate by analyzing approximately 8 embryos from their F1 progeny individually using PCR and Sanger sequencing. Germline transmission data for specific lines have been included in Supplementary Table 1.”

Supplementary Table 1. Germline targeting efficiency of base edited targets by ABE-Umax.

Gene	Code name	Tool	Germline targeting efficiency	Germline transmission rate
tyr	tyr-g2	ABE-Umax	80% (4/5)	3#, 50% (4/8)
tyr	tyr-g4	ABE-Umax	75% (3/4)	1#, 100% (8/8); 2#, 62.5% (5/8)
cdh23	cdh23-T1	ABE-Umax	66.7% (2/3)	n. a.
myo7aa	myo7aa-T3	ABE-Umax	50% (1/2)	1#, 37.5% (3/8)
myo7aa	myo7aa (Y414C)	ABE-Umax	66.7% (2/3)	1#, 37.5% (3/8); 2#, 62.5% (5/8)
adam22	adam22 (Q95R)	ABE-Umax	66.7% (2/3)	2#, 100% (8/8); 3#, 62.5% (5/8)
adam22	adam22 (S869P)	ABE-Umax	80% (4/5)	3#, 82.5% (7/8); 4#, 75% (6/8)
itsn1	itsn1-T3	ABE-Umax	100% (4/4)	n. a.

Updated supplementary table 1

9. p8 tubb4b, please mention briefly what disease this is a model for.

Response: We have edited in the results as follows: “The TUBB4B gene encodes proteins that form microtubules, essential components of the cellular skeleton. Mutations in TUBB4B have been linked to various neurological disorders that affect nervous system development and function (reference 30). A recent study identified a novel heterozygous missense variant (c.32A > G, p.Q11R) in exon 1 of TUBB4B. This variant is a potential causative mutation in patients

with early-onset hearing loss, hyperopia, hypophosphatemic rickets (HR), renal tubular Fanconi syndrome (FS), and nephrocalcinosis.”

10. Figure 3b is a nice summary, but the underlying efficiency measures at the 7 sites should also be presented in supplementary data.

Response: Thank you for your suggestion. We have added Supplementary Figure 6 to show this data.

Supplementary Figure 6. Assessment of the editing efficiency and targeting window of ABE-Umax-ex1 and ABE-Umax-ex2 using EditR to quantify A-to-G conversion frequencies at each adenine nucleotide within a 20-bp protospacer sequence. Three independent experiments were conducted and editing frequencies > 0.20 are highlighted in the heatmap. The color scale ranges from dark blue (100%) to white (0% efficiency), see color gradient represented in top right corner.

11. The color scale in 3d makes it hard to compare efficiencies - the scale in 2c is more suitable.

Response: Thank you for your suggestion. We have redrawn Figure 3d using the color scale from 2c as you suggested, but it appears rather inelegant (see Additional figure 3 for reviewer below). Alternatively, we are now using a blue color scale to present this data (se New Figure 3d below).

Additional Figure 3 for reviewer

Figure 3 d. Examining A-to-G editing efficiency of ABE-Umax, ABE-Umax-rest2, and ABE-Umax-rest1 across six endogenous genomic loci. Miseq was used to quantify A-to-G conversion frequencies at each adenine nucleotide within the 20-bp protospacer sequences. Results from three independent experiments were compiled, with editing frequencies above 20% labeled in the heatmap. Color mapping (blue to white) represents editing efficiency from 100% to 0%. Editing base positions within the gRNA are indicated by numbers.

12. The innovation of shifting the editing range from 4-8 to 3-6 or 12-15 is an important contribution, but given very short shrift. The results are only schematically presented in figure 3f and shown without quantification in Supp Fig 6. These data should be fully presented in a similar form to Figure 3b.

Response: Thank you for your appreciation. As per your suggestion, we have added this data into Figure 3e.

Figure 3 e. Evaluating the editing efficiency and targeting window of ABE-Umax-ex1-rest1 and ABE-Umax-ex2-rest1. EditR quantified A-to-G conversion frequencies at every adenine within the 20-bp protospacer. Data from three independent experiments were analyzed, with editing frequencies above 20% labeled in the heatmap. Color mapping (blue gradient to white) indicates editing efficiency from 100% to 0%. Editing base positions within the gRNA are numerically labeled.

13. p14. The base edits in *cdh23* and *tyr* both lead to cryptic splice changes. This is certainly a possible (even likely) outcome, but equally, intron retention or exon skipping could have occurred. So, the statement "We confirmed that ABE-Umax toolkit treatment resulted in the expected pre-mRNA splicing defects" could be expressed slightly differently, and the authors might explicitly mention that both changes lead to cryptic splicing. The schematics of the changes in 5g and 5h are not very clear. There are supposed to be 15 and 13 bp deletions respectively, but only 2 and 5 nucleotides are shown in the boxed area.

Response: Thank you for your suggestion. We have revised this sentence as follows: “We confirmed that ABE-Umax editing induced cryptic pre-mRNA splicing defects at mRNA level in both loci”. Also, we have updated figure 6 for clarity.

Figure 6 (g) Sanger sequencing of cDNA demonstrates *cdh23* cryptic splicing defects caused by ABE-Umax-ex2. The T-to-C substitution at the Exon 2 splicing donor site in *cdh23* induces a 15 bp deletion at the mRNA level. **(h)** Sanger sequencing reveals *tyr* splicing defects induced by ABE-Umax. The A-to-G substitution at the Exon 3 splicing acceptor site in *tyr* results in a 13 bp deletion at the mRNA level.

14. Figure 6b/c are inelegant and take some time to interpret (for example, it adds confusing clutter to add the precise sequence of both DNA strands). This detailed information should be in the supplement and the replaced in the main text by a schematized simple figure showing where the guides align on the exon with respect to the targeted base.

Response: Thank you for your suggestion. We have updated former figure 6 (now 7) as follows.

Figure 6

Figure 7. (a). Generating A-to-G conversion in a disease-relevant zebrafish model (*mitfa* c.713T>C). Protospacers are shown in black with PAM sequences in blue or green. The desired adenine for editing is highlighted in red. Potential gRNA target sites are marked with blue and green lines, respectively.

(b). Sequencing results for *mitfa* (S245P) in ABE-Umax-SpRY-induced F0 generation. The red arrowhead indicates the expected nucleotide substitution.

(c). Sequencing results for *mitfa* (S245P) in F0 generation induced by ABE-Umax-ex2 and ABE-Umax-ex2-rest1. The red arrowhead indicates the expected nucleotide substitution. Bystander

base substitutions are marked with green arrowheads in the Sanger sequencing chromatograms.

(d). Lateral view of wild-type embryos at 2 dpf with sequencing results. Scale bar: 1 mm.

(e). Lateral view of F2 homozygous embryos with the *mitfa* (S245P) mutation at 2 dpf, exhibiting pigmentation defects. Scale bar: 1 mm.

15. The experiment in Figure 6 is very helpful in that it demonstrates how an ABE-umax variant can be rationally selected. I suggest that the authors augment this by providing a summary figure, showing how the various new editors could be used for a hypothetical target along with the mean editing efficiency of each.

Response: Thank you for your suggestion. We added a schematic diagram for choosing the appropriate ABE tool at a specific site (**Figure7 f**).

Figure 7 f. Choosing the right tool for A-to-G editing depends on the presence of a specific DNA sequence motif and can be selected as follows: Look for a GG motif within 6-20 base pairs downstream of the target adenine.

Option 1: If a GG motif is present (1) For high efficiency, consider ABE-Umax (window 3-9), ABE-Umax-ex1 (window 4-12), or ABE-Umax-ex2 (window 8-16). (2) For high precision, choose ABE-Umax-rest1 (window 5-6), ABE-Umax-ex1-rest1 (window 4-6), or ABE-Umax-ex2-rest1 (window 12-15). (3) If multiple tools meet your criteria, try them all to see which one works best.

Option 2: If no GG motif is found: 1) Use ABE-Umax-SpRY. 2) Prioritize PAM sequences in this order: NGN > NAN > NYN. 3) Position the edited adenine within the 4th-8th position of the editing window.

REVIEWERS' COMMENTS

Reviewer #1 (Remarks to the Author):

The major concerns have been well addressed. While, as I previous pointed out the present title should be changed and define it as "zebrafish", otherwise addtiona cells(i.e., human cells or mice cells) should be tested. Also, the readers will be confused after reading this paper and find that it can't match the present title.

Reviewer #2 (Remarks to the Author):

The authors have done an extraordinary job revising the manuscript and addressing the comments of all the reviewers. It is now suitable for publication.

Reviewer #3 (Remarks to the Author):

The authors have thoroughly replied my previous critique. I appreciate the careful response and I particularly like the new summary figure in 7F. The whole set of experiments has been carried out to the highest standard and I predict that the toolkit will be widely adopted. I strongly recommend this for publication.

We are quite pleased to see that the reviewers appreciated the fact that we "have thoroughly replied (to) their previous critique" and "have done an extraordinary job revising the manuscript". We have made all the changes to fulfill editorial requirements.

Reviewer #1 (Remarks to the Author):

The major concerns have been well addressed. While, as I previous pointed out the present title should be changed and define it as "zebrafish", otherwise addtiona cells(i.e., human cells or mice cells) should be tested. Also, the readers will be confused after reading this paper and find that it can't match the present title.

We have changed the title. But we strongly beg to differ with Reviewer 1's request to add "zebrafish" in the title. First, the wide majority of publications in the field do not mention "in mice" or "in human cell cultures" in their titles, even when reported approaches would not work in any other eukaryotic species. Biology and gene editing are not the restricted domain of mammalian systems.

Most importantly in the context of this manuscript, we firmly assert that it will unduly restrict the scope of our study. The molecular tools we are reporting are equally applicable to other vertebrate models from fish to amphibians and mammals.

We believe the title accurately reflects the nature of our work and is more inclusive of the broader audience of Nature Communications.